# MULTI-LEVEL MULTI-TURN RL OUTPERFORMS GRPO: REASONING WITH TEXTUAL FEEDBACK

## ABSTRACT

Reinforcement learning with verifiable rewards has become the standard for training reasoning models, with Group Relative Policy Optimization (GRPO) achieving remarkable performance across mathematical, coding, and scientific domains. However, these approaches suffer from severe sample inefficiency due to sparse binary rewards, where even partially correct responses receive zero reward, providing no learning signal and causing extremely slow convergence. We propose Multi-Level Multi-Turn Reinforcement Learning (MLMT-RL), a novel framework that addresses this limitation by leveraging textual feedback to provide dense, interpretable learning signals. MLMT-RL decomposes reasoning into two synergistic levels: a higher-level policy generates task-specific contextual feedback, while a lower-level policy produces refined responses conditioned on this feedback. To ensure effective coordination between guidance generation and execution, we formulate a principled bi-level optimization framework where the higher-level policy is regularized by the lower-level value function. Additionally, we introduce novel metrics to evaluate feedback quality and utilization effectiveness. Our results demonstrate superior parameter efficiency: MLMT-RL with 2B parameters outperforms 3B GRPO models by 3.13% on MATH500, 5.18% on MBPP, and 4.77% on GPQA. Similarly, our 6B model surpasses 7B GRPO models by 3.0%, 2.8%, and 5.7% respectively. MLMT-RL thus establishes a highly efficient paradigm that delivers superior reasoning performance with significantly fewer parameters.

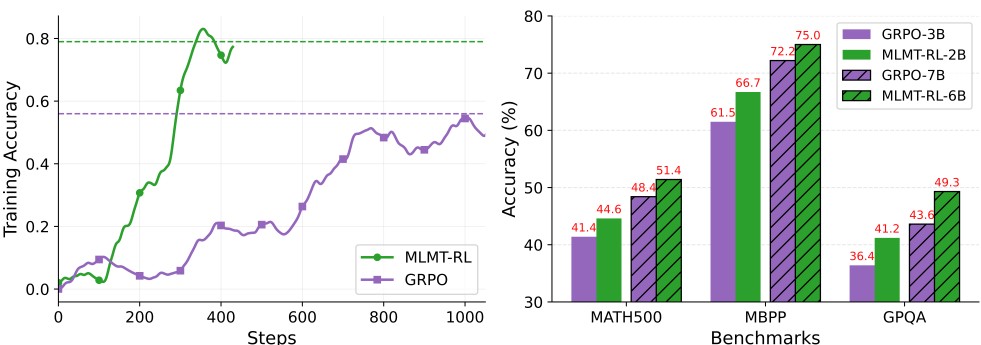

Figure 1: **MLMT-RL vs GRPO convergence rate and performance comparison across reasoning benchmarks and model scales.** (Left) On MBPP benchmark, GRPO shows significantly slow learning in sparse-reward settings, leading to sample inefficiency. By contrast, MLMT-RL's multi-level multi-turn architecture exploits higher-level feedback to enable quicker learning and better sample-efficiency (after 400 steps, GRPO achieves only 23% accuracy, whereas MLMT-RL achieves 76% accuracy). (Right) MLMT-RL outperforms GRPO-based language reasoning models across parameter scales and benchmarks, while demonstrating superior parameter efficiency. Specifically, MLMT-RL with 2B parameters outperforms GRPO with 3B parameters and MLMT-RL with 6B parameters outperforms GRPO with 7B parameters on all benchmarks. These consistent gains highlight the efficacy of MLMT-RL's hierarchical task-specific guidance over GRPO.

# 1 INTRODUCTION

Recent advances in reasoning models have demonstrated remarkable performance through reinforcement learning (RL) with verifiable rewards (Guo et al., 2025; Kumar et al., 2024b; Zhou et al., 2024). These approaches leverage ground-truth solutions to provide clear training signals in the form of sparse verifiable rewards, enabling models to solve complex mathematical, coding and scientific reasoning tasks. Despite their success, methods like Group Preference Optimization (GRPO) face a fundamental challenge: **sample inefficiency in the presence of sparse rewards**.

In GRPO, multiple candidate responses are generated for each problem and assigned binary rewards based on correctness. This creates a critical inefficiency: even partially correct responses receive zero reward, providing no learning signal. The core issue lies in the binary nature of verification-based rewards as they provide good signal when correct but offer no feedback for improvement otherwise, leading to poor sample efficiency and extremely slow convergence, as shown in Figure 1 (Left).

Prior research has shown that language models can effectively utilize natural language feedback to refine their outputs and accelerate learning Scheurer et al. (2024; 2022); Pan et al. (2023). Thus, a promising solution is to incorporate **textual feedback** into the learning process. Unlike binary rewards, textual feedback provides dense, interpretable signals that can help models understand not just their response correctness, but specifically how to improve. As an analogy, experienced coaches guide athletes by providing specific, actionable feedback like "adjust your follow-through" or "keep your eyes on target", rather than only rewarding on hitting the target or penalizing when they miss.

Based on this insight, we propose employing an auxiliary language model to generate task-specific feedback, thus providing denser learning signals for reasoning tasks. Our framework operates through a multi-turn interaction: a primary language model generates an initial response, an auxiliary model provides targeted feedback based on the response, and the primary model refines its response based on the feedback. This introduces a *multi-level multi-turn* approach: the higher-level model provides task-specific feedback and the lower-level model generates refined responses based on the feedback.

However, this multi-level approach raises two critical questions: $(i)$ how to generate optimal feedback, and $(ii)$ how to learn effectively from this feedback. To generate optimal feedback, we investigate three feedback paradigms: (1) fixed, task-agnostic feedback (Kumar et al., 2024b), (2) feedback from pre-trained models without task-specific fine-tuning, and (3) task-specific feedback from a language model trained to maximize verifiable rewards. Our empirical analysis shows that task-specific trained feedback significantly outperforms the alternatives across all benchmarks.

To learn effectively from feedback, we develop a principled bi-level optimization framework where the higher-level model learns to generate effective feedback conditioned on the lower-level model's current capabilities, while the lower-level model learns to incorporate this feedback to generate refined responses. In order to quantitatively measure feedback effectiveness, we introduce the following metrics: $(i)$ *feedback optimality metric*, which measures the quality of generated feedback using an LLM as a judge, and two *feedback compatibility metrics*, which evaluate how well the lower-level model incorporates feedback into refined responses by computing $(i)$ accuracy increase, and $(ii)$ increase in the percentage of correct responses due to feedback.

**Remark**. We emphasize that MLMT-RL is not positioned as a competitor to SOTA reasoning models that rely on extensive multi-phase training, large-scale supervised fine-tuning, or massive teacher distillation (Guo et al., 2025). Instead, MLMT-RL offers a complementary paradigm that achieves efficiency and robustness through multi-level multi-turn RL with modest resources.

Our main contributions are:
**1. Multi-Level Multi-Turn RL framework:** We propose MLMT-RL, a novel bi-level approach that addresses limitations of prior reasoning models by decomposing reasoning into two synergistic levels: higher-level feedback generation and lower-level refinement based on this feedback.
**2. Novel evaluation metrics for evaluating feedback** We introduce multiple metrics: *feedback optimality metric* to assess the quality of generated feedback, and three *feedback compatibility metrics* to measure how effectively the lower-level policy is able to improve based on provided feedback.
**3. Empirical analysis:** We show that 2B parameter MLMT-RL models outperform larger 3B GRPO-based models by 3.13% on MATH500, 5.18% on MBPP, and 4.77% on GPQA, and 6B parameter MLMT-RL models surpass larger 7B GRPO models by 3%, 2.8%, and 5.7% on the benchmarks, showing superior reasoning performance with fewer parameters (Figure 1 (Right)).

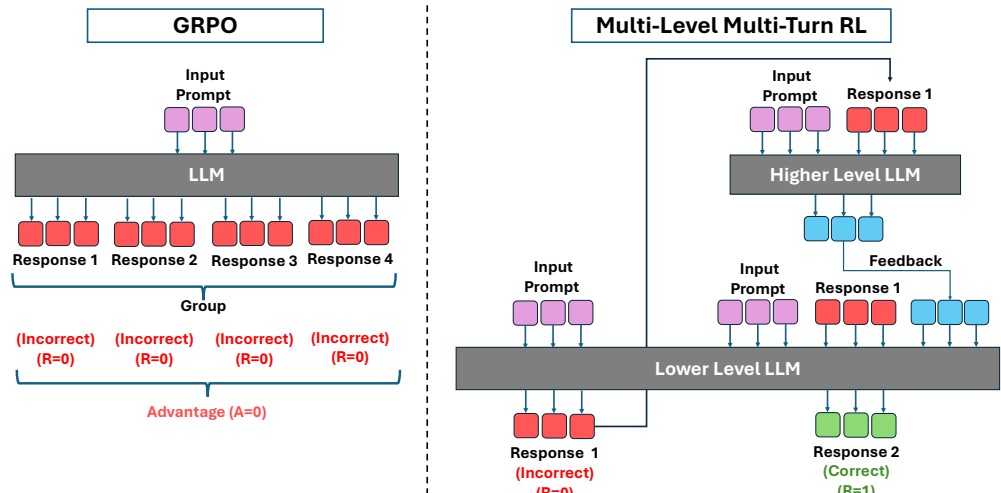

Figure 2: **MLMT-RL Overview:** (**Left**) In GRPO, the model generates a group of responses (e.g., group size = 4) based on the input prompt. In sparse reward scenarios, where most or all responses yield zero reward (R=0), the advantage becomes zero, resulting in a non-existent learning signal and consequently slow learning. (**Right**) In MLMT-RL, the higher-level LLM generates task-specific feedback conditioned on the input prompt and the lower-level policy's initial response. The lower-level policy then uses this feedback to refine its output and produce the correct response (R=1).

## 2    RELATED WORK

**LLM Reasoning and Self-Correction.** Recent advances in language reasoning have been driven by GRPO models (Shao et al., 2024), which show strong capabilities (Guo et al., 2025), but suffer from slow convergence and sample inefficiency in the presence of sparse rewards. Self-correction based variants use environmental feedback (e.g., code execution (Jain et al., 2024), tool interactions (Chen et al., 2023)) and intrinsic approaches without external signals (Kamoi et al., 2024; Huang et al., 2023). While zero-shot prompting often degrades performance (Huang et al., 2023; Zheng et al., 2024), supervised methods with human corrections (Saunders et al., 2022) or model distillation (Ye et al., 2023) show promise but require substantial supervision.

**Multi-Turn Reinforcement Learning.** Multi-turn RL enables efficient reasoning without external supervision. Early methods include value-based approaches (Zhou et al., 2024; Shani et al., 2024) for correction quality estimation and policy-based techniques (Shao et al., 2024) for direct optimization. SCoRe by Kumar et al. (2024a) trains on self-generated trajectories, outperforming supervised baselines. However, existing methods use static, task-agnostic guidance that lacks problem-specific nuance for diverse tasks. MLMT-RL addresses this through multi-level optimization for dynamic, task-specific guidance, while mitigating rank bias and enabling targeted credit assignment.

**Hierarchical Reinforcement Learning (HRL).** HRL improves efficiency via temporal abstraction (Nachum et al., 2019), decomposing tasks into subtasks (Sutton et al., 1999; Barto & Mahadevan, 2003). Vanilla HRL faces non-stationarity from evolving lower-level policies (Nachum et al., 2018; Levy et al., 2018). Recent solutions include optimal behavior simulation (Levy et al., 2018), experience relabeling (Nachum et al., 2018), and bi-level formulations (Singh et al., 2024). MLMT-RL uses bi-level optimization to manage non-stationarity and promote reasoning diversity.

## 3    PRELIMINARIES

**Group Relative Policy Optimization.**

GRPO trains reasoning models by generating groups of candidate responses, and optimizing over the corresponding binary verifiable rewards. For input problem $x$, the model generates $G$ responses $\{y_i\}_{i=1}^G \sim \pi_\theta(\cdot|x)$ and receives rewards $\mathcal{R}(y_i, y_i^*)$ returning 1 for correct solutions and 0 otherwise.

---

**MATH Arithmetic Example**

**Prompt:** If the domain of the function $\log x^2$ is $x < a$ or $x > b$, for some $a$ and $b$, find $a + b$.

> **GRPO**
>
> **Response 1 (Incorrect):**
> ... Apply the rule $\log(x^2) = 2\log(x)$ to all $x$, so domain is $(-\infty, \infty)$.
> **Response 2 (Incorrect):**
> ... Treat $\log(x^2)$ as valid for all real $x$, giving $a = -\infty, b = \infty$.
> **Response 3 (Incorrect):**
> ... Restrict to negative side, so $a = -\infty, b = 0$ and $a + b = -\infty$.
> **Response 4 (Incorrect):**
> ... Approximate solution by guessing $a = 1, b = 2$ so $a + b = 3$.

> **MLMT-RL**
>
> **First Response (Incorrect):**
> In this attempt, the solver again applies the power rule, restricts $x > 0$, assigns $a = 0$ and $b = \infty$, and reports $a + b = \infty$.
>
> > **Task-Specific Feedback**
> >
> > Rewrite the function using the power rule and ensure its argument is positive. Consider all real $x$ satisfying $x^2 > 0$, including negatives. Express the domain as $x < a$ or $x > b$ and compute the sum of those endpoints.
>
> **Improved Response (Correct):**
> By noting that $x^2 > 0$ for all $x \neq 0$, the domain splits into $(-\infty, 0) \cup (0, \infty)$, so $a = 0$ and $b = 0$, giving $a + b = 0$.

Figure 3: **Arithmetic Illustrative Example: MLMT-RL vs GRPO Comparison.** For the arithmetic example, GRPO generates $G = 4$ responses but each of them fail to solve this arithmetic problem, leading to sparse rewards $R_i = 0 \ \forall i \in [1, 4]$. In contrast, MLMT-RL's higher-level policy generates task-specific, context-aware feedback that enables the lower-level policy to successfully generate correct response and solve the task, thereby demonstrating superior reasoning refinement and overall performance.

**Objective.** The GRPO objective can be formulated as $J_{\text{GRPO}}(\theta) = \mathbb{E}_{y \sim \pi_\theta(\cdot|x)}[R(x, y) - \hat{R}]$. where $R(x, y)$ is the final task reward and $\hat{R}$ is a group-normalized baseline computed by $\hat{R} = \frac{1}{G}\sum_{i=1}^{G} R_i$. The group-normalized advantage for each sequence is computed as $A_i = \frac{R_i - \hat{R}}{\sigma_R + \epsilon}$, where $\sigma_R$ is the standard deviation of rewards within the group and $\epsilon$ is the numerical stability constant.

**Limitation.** GRPO's reliance on binary verification rewards creates a fundamental bottleneck: when all responses in a group are incorrect, the resulting zero advantages eliminate learning signals entirely. This advantage collapse occurs because GRPO computes relative rewards within each group: if every response receives $R_i = 0$, then reward $\hat{R} = 0$ and advantage $A_i = 0 \forall i$, providing no learning signal. We demonstrate this limitation through comprehensive empirical analysis:

**1. Learning stagnation.** Figure 1 (Left) shows that on MBPP task using LLAMA-3.2-1B backbone, GRPO shows minimal performance improvement for around 600 steps, achieving only 23% accuracy compared to MLMT-RL's 79% at 600 step count. This learning stagnation occurs because GRPO's updates become increasingly sparse as training progresses, due to sparse reward signals.

**2. Advantage collapse Analysis.** We empirically measure the *Advantage Collapse frequency* by finding the percentage of groups during training that lead to advantage $A = 0$. On the MBPP task, we observe the advantage collapse frequency to be 74%, which implies that 74% of the training groups experience complete advantage collapse, leading to most training steps providing no gradient updates.

**3. Zero reward frequency.** We also measure the *Reward Collapse frequency*, by measuring the percentage of responses across training that lead to rewards $R = 0$. We observe that on MBPP task, the reward collapse frequency is 83%, implying that 83% of the trajectories lead to 0 rewards, further confirming the prevalence of extremely sparse learning signals in GRPO.

**4. Illustrative failure case.** We provide an illustrative example in Figure 3. With group size

$G = 4$, GRPO generates four responses that all fail to solve the task, and thus each receive sparse reward of $R_i = 0$, implying advantage $A = 0$. In contrast, although MLMT-RL fails to solve the task in the first response, the model is able to successfully leverage the higher-level LLM targeted feedback to generate the correct response in the second turn. We provide more illustrative examples in Appendix 7.9.

The above limitations and empirical analysis demonstrates the need for denser, task-specific feedback signals, unlike binary rewards that provide only sparse feedback. Based on prior research, we note that textual feedback can be a richer alternative that can identify specific errors in prior attempts and suggest denser and task-specific correction feedback.

## 4 PROPOSED METHODOLOGY

We propose Multi-Level Multi-Turn Reinforcement Learning (MLMT-RL), a hierarchical framework that addresses GRPO's sparse reward limitation by leveraging textual feedback to provide dense, interpretable learning signals. MLMT-RL decomposes reasoning into two synergistic levels: a higher-level policy that generates task-specific feedback and a lower-level policy that produces refined responses conditioned on this feedback. We now explain our framework in detail.

### 4.1 MULTI-LEVEL MULTI-TURN FRAMEWORK

Given an input problem $x$, the lower-level policy $\pi_\theta^L$ first generates an initial attempt $z \sim \pi_\theta^L(\cdot \mid x)$ in the first turn. This stage resembles prior RL-based approaches that rely on verifiable sparse rewards. However, during initial training stages when models are untrained, most responses are incorrect, leading to zero rewards ($R = 0$). Specifically, in the GRPO approach, when all responses in the group are incorrect ($R_i = 0, \forall i \in [1, G]$), the advantage $A = 0$ (Figure 2 (Left)), resulting in non-existent learning signals and extremely slow learning.

Our framework deals with this issue by leveraging a higher-level policy model $\pi_\phi^H(\cdot \mid x)$ to generate context-aware feedback as follows: the higher-level policy conditioned on the input prompt $x$ and the initial attempt $z$, generates a feedback $g \sim \pi_\phi^H(\cdot \mid x, z)$ that acts as a denser feedback signal describing how to generate corrected response based on the initial attempt $z$ (Figure 2 (Right)). Consequently, the lower-level policy leverages this feedback to generate a refined response $\hat{y} \sim \pi_\theta^L(\cdot \mid x, z, g)$ in the second turn, thereby addressing the limitations of prior approaches.

In order to efficiently solve the above multi-level multi-turn problem, we design a hierarchical MDP formulation that formalizes the reasoning process as a hierarchical Markov Decision Process (MDP).

### 4.2 HIERARCHICAL MDP FORMULATION

Given an input dataset $\mathcal{D} = \{(x_i, y_i^*)\}_{i=1}^N$, where $x_i$ are input problems and $y_i^*$ are target responses, we formalize the reasoning process as a hierarchical MDP. In this framework, a policy iteratively refines its responses over multiple turns in a multi-turn setting with $L$ turns (we use $L = 2$ in our case, although it can be extended to larger $L$ values). We now define the hierarchical MDPs as follows.

**Higher-Level MDP.** We represent the higher-level MDP as $M^H = (S^H, A^H, P^H, R^H)$, where $S^H$ is the state space, $A^H$ is the action space, $P^H$ is the state transition probability function, and $R^H$ is the reward function. The higher-level policy is denoted as $\pi_\phi^H$ with parameters $\phi$, which generates feedback $g \sim \pi_\phi^H(\cdot \mid x, z)$ conditioned on the input $x$ and the previous attempt $z$.

**Lower-Level MDP.** We represent the lower-level MDP as $M^L = (S^L, A^L, P^L, R^L)$, where $S^L$ is the state space, $A^L$ is the action space, $P^L$ is the state transition probability function, and $R^L$ is the reward function (typically a sparse binary reward $R(\hat{y}, y^*) \in \{0, 1\}$ based on correctness verification). The lower-level policy is denoted as $\pi_\theta^L$ with parameters $\theta$, which generates outputs $\hat{y} \sim \pi_\theta^L(\cdot \mid x, z, g)$ conditioned on the input $x$, the previous attempt $z$, and the higher-level feedback $g$. The lower-level policy's value function represents the expected cumulative reward from a given state conditioned on the higher-level feedback $g$: $V_{\pi_\theta^L}^L(x, g) = \mathbb{E}_{\hat{y} \sim \pi_\theta^L}[Q_{\pi_\theta^L}^L(x, g, \hat{y})]$, where $Q_{\pi_\theta^L}^L(x, g, \hat{y})$ is the action-value function estimating the expected reward after following policy $\pi_\theta^L$.

In MLMT-RL, the overall objective is to maximize the expected final rewards, which is:

$$\mathcal{J}(\theta, \phi) = \mathbb{E}_{(x,y^*)\sim D,\ z\sim\pi_\theta^L(\cdot|x),\ g\sim\pi_\phi^H(\cdot|x,z),\ \hat{y}\sim\pi_\theta^L(\cdot|x,z,g)}\left[R(\hat{y}, y^*)\right]. \tag{1}$$

This objective maximizes the expected task reward $R(\hat{y}, y^*)$ based on the second-turn predicted response $\hat{y}$ and target response $y^*$. The expectation is over: (i) the input $x$ and target response $y^*$ sampled from the dataset $D$, (ii) the first response $z$ sampled from the lower-level policy $\pi_\theta^L(\cdot \mid x)$ conditioned on the input $x$, (iii) the feedback $g$ sampled from the higher-level policy $\pi_\phi^H(\cdot \mid x, z)$ conditioned on the input $x$ and first-turn attempt $z$, and (iv) the final response $\hat{y}$ sampled from the lower-level policy $\pi_\theta^L(\cdot \mid x, z, g)$ in the second turn. Both the hierarchical policies leverage this objective to learn their parameters $\phi$ and $\theta$ via RL (e.g., REINFORCE Williams (1992)).

## 4.3 BI-LEVEL FORMULATION

Although we have outlined the multi-level training objectives using the hierarchical MDP formulation, there exists an inherent inter-dependency between the two hierarchical levels. The higher-level policy $\pi_\phi^H$ generates feedback $g$ which is provided to the lower-level policy $\pi_\theta^L$ to condition its generation of the refined output $\hat{y}$. In turn, the lower-level policy's output determines the final reward, which is used to train the higher-level policy. This inter-dependency calls for a principled formulation which should enable: ($i$) *feedback optimality*: where the higher level should generate optimal feedback according to the capabilities of the lower level policy, and ($ii$) *feedback compatibility*: where the lower-level policy should be able to effectively leverage this feedback to generate refined responses.

To develop a principled approach that resolves this inter-dependency, we formalize the problem as a bi-level optimization problem following Singh et al. (2024). Let $\mathcal{J}_H(\pi^H, \pi_*^L(\pi^H))$ represent the higher-level objective and $\mathcal{J}_L(\pi^L|\pi^H)$ represent the lower-level objective (we drop the parameters $\phi$ and $\theta$ for ease of representation). The bi-level formulation can be represented as:

$$\max_{\pi^H} \mathcal{J}_H(\pi^H, \pi_*^L(\pi^H)) \quad \text{s.t.} \quad \pi_*^L(\pi^H) = \arg\max_{\pi^L} \mathcal{J}_L(\pi^L|\pi^H) = \arg\max_{\pi^L} V^L(\pi^H), \tag{2}$$

where $\pi_*^L(\pi^H)$ represents the optimal lower-level policy given the higher-level policy $\pi^H$, and $V^L(\pi^H)$ is the lower level value function, and the optimal lower policy maximizes the lower level value function. Thus, the higher-level objective is constrained by lower-level policy optimality. Using the recent advancements in the optimization literature (Liu et al., 2022), we can show that Equation 2 can be utilized to derive the following objective for the higher-level policy:

$$\mathcal{J}_\phi^H = \mathbb{E}_{(x,y^*)\sim D,\ z\sim\pi_\theta^L(\cdot|x),\ g\sim\pi_\phi^H(\cdot|x,z),\ \hat{y}\sim\pi_\theta^L(\cdot|x,z,g)}\left[R(\hat{y}, y^*) + \lambda(V^L(x,g) - V_*^L(x,g)\right], \tag{3}$$

where $V^L(s,z)$ and $V_*^L(s,z)$ are the current and optimal lower-level value functions conditioned on $x$ and $z$, and $\lambda$ is the regularization parameter that controls the trade-off between immediate reward maximization and value function regularization. The first term, $R(\hat{y}, y^*)$, rewards the higher-level for selecting feedback $g$ that enables the lower-level to produce outcomes $\hat{y}$ matching the targets $y^*$. The regularization term, $\lambda(V^L(x,g) - V_*^L(x,g))$ (where $\lambda \geq 0$), encourages the higher-level to generate feedback that steers the lower-level policy toward near-optimal behavior, thereby both preventing degenerate solutions (such as the lower-level ignoring the feedback) and facilitating effective alignment between the two levels. We provide the complete derivation of Equation 3 in Appendix 7.1. This formulation ensures that the higher-level policy generates optimal feedback that is well-aligned with the lower-level policy's current capabilities.

**Higher-Level Gradient.** The gradient of our upper-level objective $\mathcal{J}_\phi^H$ with respect to $\phi$ is:

$$\nabla_\phi \mathcal{J}_\phi^H = \mathbb{E}_{x,y^*,z,g,\hat{y}}\left[\nabla_\phi \log \pi_\phi^H(g \mid x,z) \cdot \left(R(\hat{y}, y^*) + \lambda(V^L(x,g) - V_*^L(x,g))\right)\right], \tag{4}$$

where the expectation is over $(x, y^*) \sim D, z \sim \pi_\theta^L(\cdot \mid x), g \sim \pi_\phi^H(\cdot \mid x, z), \hat{y} \sim \pi_\theta^L(\cdot \mid x, z, g)$. The full derivation is provided in Appendix 7.2.

**Lower-Level Gradient.** The gradient of our lower-level objective $\mathcal{J}_\phi^H$ with respect to $\theta$ is:

$$\nabla_\theta J_\phi^H = \mathbb{E}_{x,y^*,z,g,\hat{y}}\left[\nabla_\theta \log \pi_\theta^L(z \mid x) \cdot \tilde{R} + \nabla_\theta \log \pi_\theta^L(\hat{y} \mid x,z,g) \cdot \tilde{R}\right] +$$
$$\lambda \mathbb{E}_{x,y^*,z,g}\mathbb{E}_{\hat{y}'}\left[\nabla_\theta \log \pi_\theta^L(\hat{y}' \mid x,z,g) \cdot R(\hat{y}', y^*)\right], \tag{5}$$

Figure 4: **(Left) Comparison of MLMT-RL and GRPO on the MBPP benchmark**. GRPO exhibits extremely slow learning in sparse-reward environments, resulting in sample inefficiency. In contrast, MLMT-RL's multi-level multi-turn framework leverages higher-level feedback for rapid learning and faster, more sample-efficient convergence. **(Right) MLMT-RL versus GRPO performance comparison across scales.** MLMT-RL with 2B parameters (LLaMA-3.2-1B at each higher and lower level) outperforms GRPO-trained models with 3B parameters, while MLMT-RL with 6B parameters (LLaMA-3.2-3B at each higher and lower level) exceeds those with 7B parameters, demonstrating superior parameter efficiency and overall performance.

where the expectation $\mathbb{E}_{(x,y^*,z,g,\hat{y})}$ is over $(x, y^*) \sim D, z \sim \pi_\theta^L(\cdot \mid x), g \sim \pi_\phi^H(\cdot \mid x, z), \hat{y} \sim \pi_\theta^L(\cdot \mid x, z, g)$, and $\tilde{R} = R(\hat{y}, y^*) + \lambda \left( V_L(x, g) - V_L^*(x, g) \right)$. The full derivation is provided in Appendix 7.3. This is estimated by Monte-Carlo sampling and used to update parameters $\phi$ and $\theta$ using standard policy gradient. The gradients are computed via automatic differentiation.

**Practical Approximation.** The objective in Equation 3 requires computing the optimal value function $V_*^L(\pi^H)$ which is often intractable. To enable practical implementation, we approximate the optimal value function $V_*^L(\pi^H)$ with the value function $V_k^L(\pi^H)$, which we get by taking $k$ gradient steps to learn the parameters of $V_k^L(\pi^H)$ for every higher-level policy gradient step. We empirically found that this approximation results in efficient implementation and is well justified in practice for even modest values of $k$. We provide the detailed algorithm in Appendix 7.4.

# 5 EXPERIMENTS

Our empirical analysis addresses the following core research questions to validate our contributions:
1. Does MLMT-RL outperform GRPO-based reasoning models in terms of performance, parameter efficiency, and sample efficiency?
2. How does MLMT-RL compare to state-of-the-art reasoning methods across various model scales?
3. In MLMT-RL, does the higher-level policy generate optimal feedback, and can the lower-level policy effectively utilize this feedback?
4. Can use a single model for both feedback generation and response refinement in MLMT-RL?

**Benchmarks.** We evaluate on three domains:
1. MATH-500: a 500-problem subset of the MATH corpus for mathematical reasoning.
2. MBPP: on 974 Python tasks scored with HumanEval for code generation.
3. GPQA: using 3000 graduate-level questions from PHYSICS dataset for scientific reasoning.
The experiments are performed on LLaMA-3.2 (1B, 3B, and 8B) and Qwen-2 (1.5B, 7B) models. We provide the training efficiency details for MLMT-RL and the baselines in Appendix 7.5 and rigorous details on experiments and evaluations prompts in the Appendix 7.6. The correction instruction prompt templates for the datasets are provided in Appendix 7.7.

**1. Does MLMT-RL outperform GRPO-based reasoning models in terms of performance, parameter efficiency, and sample efficiency?**

**Convergence speed.** In Figure 4, we show that MLMT-RL outperforms larger GRPO models with superior sample efficiency, parameter efficiency, and faster convergence. As shown in the left panel, GRPO improves minimally until around 600 steps on the MBPP benchmark, reaching only 22% accuracy before gradual gains. In contrast, MLMT-RL achieves 75% accuracy in just 400 steps due to its task-specific feedback, providing dense signals for faster learning and better sample efficiency. **Performance across model scales.** To assess parameter efficiency, we compare smaller MLMT-RL

| Model | 1B (LLaMA-3.2–1B) | | | 3B (LLaMA-3.2–3B) | | | 8B (LLaMA-3.1–8B) | | |
|---|---|---|---|---|---|---|---|---|---|
| Method | MATH | MBPP | GPQA | MATH | MBPP | GPQA | MATH | MBPP | GPQA |
| BC | 26.20 | 50.00 | 30.20 | 42.40 | 68.50 | 37.10 | 53.0 | 74.5 | 39.0 |
| ZSCoT | 25.80 | 41.29 | 26.80 | 41.30 | 60.98 | 32.70 | 50.9 | 72.8 | 34.2 |
| ArCHer | 30.60 | 52.44 | 33.60 | 43.60 | 69.50 | 40.90 | 55.5 | 76.0 | 43.0 |
| SCoRe | 34.80 | 57.32 | 34.40 | 45.20 | 70.73 | 42.60 | 56.8 | 77.3 | 44.6 |
| MLMT-RL | **44.56** | **66.73** | **41.20** | **51.40** | **75.00** | **49.30** | **60.2** | **79.8** | **56.3** |

Table 1: In this table, we compare MLMT-RL against various SOTA baselines across model sizes (1B, 3B and 8B). MLMT-RL consistently outperforms single-turn (ZSCoT, BC) and multi-turn (ArCHer, SCoRe) methods on MATH, MBPP, and GPQA across LLaMA backbones of 1B, 3B and 8B size models.

models against larger GRPO models (2B MLMT-RL vs 3B GRPO, and 6B MLMT-RL vs 7B GRPO). In both cases, MLMT-RL outperforms GRPO, highlighting its parameter efficiency. As shown in the right panel of Figure 4, the 2B MLMT-RL beats the 3B GRPO by 3.13 points on MATH, 5.18 on MBPP, and 4.77 on GPQA. The 6B MLMT-RL exceeds the 7B GRPO by 3.0, 2.8, and 5.7 points respectively. These results show MLMT-RL exhibits stronger reasoning performance with fewer parameters and faster convergence, making it a more efficient alternative to GRPO.

## 2. How does MLMT-RL compare to state-of-the-art (SOTA) reasoning approaches across different model scales?

We compare MLMT-RL to prior reasoning methods to assess its design efficacy.

**Baselines.** To isolate the specific contributions of MLMT-RL's multi-level multi-turn structure and task-specific learned guidance, we compare against baselines that vary in these elements, allowing us to quantify the performance gains from each component. These include: (i) *Behavioral Cloning (BC)* (Torabi et al., 2018), a supervised fine-tuning method using expert trajectories without RL or multi-turn elements, which allows us to highlight the benefits of using reinforcement learning training with verifiable rewards; (ii) *Zero-Shot Chain-of-Thought (ZSCoT)*, a two-turn multi-level approach that we implemented using a pretrained LLM without fine-tuning at each level, where the higher-level model provides feedback not optimized using verifiable rewards, thus allowing us to evaluate the impact of multi-level reward-based fine-tuning; (iii) *ArCHer* (Zhou et al., 2024), a single-turn RL approach with separate critic and actor networks but no multi-turn feedback, to assess the value added by our multi-turn interaction; and (iv) *SCoRe* (Kumar et al., 2024a), a multi-turn RL method that fine-tunes the lower-level policy using verifiable rewards but relies on fixed, task-agnostic feedback, to demonstrate the advantages of learning task-specific feedback over fixed guidance.

**Analysis.** Table 1 shows results across 1B, 3B, and 8B model scales on MATH, MBPP, and GPQA benchmarks. Although single-turn approach BC outperforms the ZSCoT baseline, ArCHer outperforms BC, demonstrating that this RL-based method leverages reward signals for more efficient task solving than BC and ZSCoT across all benchmarks and scales. SCoRe surpasses BC, ZSCoT, and ArCHer, validating the effectiveness of its multi-turn approach. However, MLMT-RL consistently outperforms all baselines, including SCoRe across model scales, including a substantial advance on the challenging GPQA dataset at 8B scale (from 44.6% to 56.3%). As model capacity increases, all methods improve due to greater base capabilities, but MLMT-RL gains more from its multi-level structure. These results show that across domains, MLMT-RL is an effective alternative, particularly in resource-constrained settings where parameter and sample efficiency are critical.

## 3. In MLMT-RL, does the higher-level policy generate optimal feedback, and can the lower-level policy effectively utilize this feedback?

Our proposed MLMT-RL framework relies on higher-level feedback to refine lower-level responses, where the performance depends on two key factors: the higher-level policy's ability to generate high-quality, task-relevant feedback, and the lower-level policy's capacity to incorporate this feedback for improved accuracy. To critically evaluate this, we introduce targeted metrics to quantify feedback optimality and compatibility, and compare MLMT-RL against carefully designed baselines that ablate whether higher-level and lower-level policies are fine-tuned on verifiable rewards.

**Feedback optimality and feedback compatibility metrics.** We define novel metrics to quantitatively assess feedback generation and utilization in MLMT-RL: ($i$) in the *feedback optimality* (FO) metric,

| Method | Higher-level fine-tuned | Lower-level fine-tuned | Accuracy | FO | $\Delta_{\text{acc}}(t_1, t_2)$ | $\Delta_{i \to c}(t_1, t_2)$ |
|---|---|---|---|---|---|---|
| ZSCoT | No | No | 35.40 % | 2.1 | +9.60 % | +12.20 % |
| SCoRe-FF | No | Yes | 39.60 % | 2.1 | +7.40 % | +17.80 % |
| TF-LBM | Yes | No | 38.00 % | 3.0 | +8.80 % | +16.00 % |
| MLMT-RL w/o $V_{\text{reg}}$ | Yes | Yes | 41.80 % | 3.6 | +9.60 % | +19.20 % |
| MLMT-RL | Yes | Yes | **44.20 %** | **4.2** | **+11.40 %** | **+21.20 %** |

Table 2: **Feedback optimality and compatibility analysis** We conduct evaluations on MATH-500 with LLAMA-3.2-1B for various baselines that differ by whether the hierarchical levels are RL-fine-tuned. We report the accuracy, feedback optimality metric (FO), and two feedback compatibility metrics: $(i)$ accuracy increase from turn 1 to turn 2 $\Delta_{\text{acc}}(t_1, t_2)$, and $(ii)$ the fraction flipped from incorrect to correct $\Delta_{i \to c}(t_1, t_2)$. MLMT-RL achieves higher accuracy and superior feedback optimality and feedback compatibility against baselines.

an LLM judge scores the feedback on a 1–5 scale based on its precision, relevance, and actionability with respect to the input query and first-turn response. Here, the outputs of all methods are passed together to enforce relative grading (see Appendix 7.7 for the full evaluation prompt). We also define two *feedback compatibility* metrics, where we compare via: $(ii)$ accuracy increase from turn 1 (initial response) to turn 2 (refined response) ($\Delta_{acc}(t_1, t_2)$), and $(iii)$ increase in fraction of incorrect $\to$ correct responses ($\Delta_{i \to c}(t_1, t_2)$). These metrics evaluate whether a method is able to generate high-quality feedback and leverage it effectively to refine its responses.

**Baselines.** We compare MLMT-RL against multiple baselines that differ by whether higher and lower level policies are RL fine-tuned on verifiable rewards: $(i)$ HZCoT (Hierarchical Zero-Shot CoT), where both levels are pre-trained models without RL fine-tuning; $(ii)$ SCoRe-FF, a multi-turn baseline with an RL-fine-tuned lower level but fixed, task-agnostic and generic feedback (e.g., "Please think carefully and generate a correct output."); $(iii)$ TF-LBM (Fine-tuned feedback Lower-Base Model), where the higher-level policy is RL-trained as in MLMT-RL but the lower level is a pre-trained model; and $(iv)$ MLMT-RL w/o $V_{\text{reg}}$, which omits value function regularization from the bi-level objective. We use this baseline to analyze the importance of bi-level optimization in our framework.

**Analysis.** In Table 2, we compare accuracy, feedback optimality (FO), and feedback compatibility metrics for all baselines. As seen from the table, MLMT-RL consistently achieves the highest accuracies, outperforming baselines and demonstrates superior feedback generation and utilization for refinement. The baselines without higher-level RL fine-tuning, like HZCoT and SCoRe-FF, show low FO scores, indicating that pre-trained or fixed feedback lacks the task-specific precision needed for optimal feedback, leading to modest compatibility gains. In contrast, TF-LBM with fine-tuned higher-level but pre-trained lower-level policies improves FO but still shows limited compatibility metric values, emphasizing the need for lower-level as well fine-tuning for effective feedback compatibility. Our variant without value regularization (MLMT-RL w/o $V_{\text{reg}}$) outperforms earlier baselines but lags behind full MLMT-RL, underscoring the importance of our bi-level optimization in coordinating inter-level dependencies. These results confirm that fine-tuning both levels using our bi-level framework enables MLMT-RL to overcome sparse-reward limitations via dense, actionable feedback, yielding consistent improvements. We also analyze convergence in Figure 4, showing MLMT-RL's faster learning rate and superior sample efficiency among these baselines.

### 4. Can MLMT-RL use a single model for both feedback generation and response refinement?

So far, we have discussed MLMT-RL that employs two separate models: one for generating task-specific feedback and another for refining responses conditioned on feedback. Now, we ask the question: can a single model can handle both tasks? This effectively creates a *self-critiquing* framework that generates an initial response, generates feedback on it, and then refines its response based on the feedback. Note that this is different from prior thinking tokens or single-level multi-turn based approaches, which do not include a feedback generation step on prior responses for targeted improvements. This single-level variant also minimizes the computational costs compared to two models framework. We implemented and tested a single-model variant of MLMT-RL using the LLaMA-3.2-1B backbone on MATH-500 and MBPP benchmarks. Table 3 shows that the single-model (1M) variant achieves performance comparable to the original

| Method | MATH-500 | MBPP |
|---|---|---|
| MLMT-RL (2M) | 44.56 | 66.73 |
| MLMT-RL (1M) | 44.84 | 65.24 |

Table 3: Comparing MLMT-RL variants: (2M) two models vs. (1M) single model.

two-model (2M) variant, which shows that a single model can effectively generate self-feedback and refine responses, opening future avenues for self-critiquing models. This variant reduces compute costs and simplifies deployment without any performance loss, enhancing MLMT-RL's practicality. **Heterogeneous Model Pairing Analysis.** We also examine whether MLMT-RL's performance gains stem primarily from using same-family models for the hierarchical policies, or from the framework's multi-turn multi-level interaction that addresses sparse reward issues, in Appendix 7.8.

## 6 DISCUSSION

**Limitations.** MLMT-RL's bi-level optimization requires accurate value estimation at both levels, where approximations may lead to bias. While current experiments focus on mathematical reasoning, code generation, and scientific reasoning, we would also like to test adaptability to open-ended language or multimodal tasks in future, while improving computational efficiency.

**Conclusion.** In this work, we introduced MLMT-RL, a hierarchical framework that enhances reasoning capabilities in language models, and addresses the sample inefficiency issue of existing methods like GRPO in sparse reward scenarios. By proposing a bi-level framework for efficiently decomposing the reasoning process into synergistic higher-level feedback generation and lower-level response refinement, MLMT-RL provides dense, task-specific learning signals that enable faster convergence and superior performance across mathematical, coding, and scientific domains. Our empirical results demonstrate that MLMT-RL achieves remarkable parameter efficiency, with smaller models outperforming larger GRPO counterparts by substantial margins. Furthermore, novel metrics for feedback optimality and compatibility validate the framework's effectiveness in generating and utilizing guidance. This work opens promising directions for multi-level multi-turn approaches to language model reasoning, with future extensions to broader domains and enhanced interpretability.

## ETHICS STATEMENT

This work introduces MLMT-RL, a multi-level multi-turn framework for improving reasoning capabilities in language models. We acknowledge several ethical considerations. First, enhanced reasoning capabilities in language models could potentially be misused for generating misleading or harmful content, though our focus on mathematical, coding, and scientific reasoning tasks limits immediate risks. Second, our training methodology requires computational resources (16.5 hours of training time as reported), contributing to environmental impact through energy consumption. We encourage responsible use of computational resources and consideration of energy-efficient alternatives where possible. Third, our evaluation relies on existing datasets (MATH-500, MBPP, GPQA) and we respect their original licensing and usage terms. The hierarchical feedback mechanism in our approach does not introduce new privacy concerns beyond standard language model training. We emphasize that our work aims to improve sample efficiency and parameter efficiency in reasoning tasks, potentially reducing overall computational requirements for achieving similar performance levels. Users of this technology should consider potential downstream applications and ensure responsible deployment in accordance with ethical AI principles.

## REPRODUCIBILITY STATEMENT

To ensure reproducibility of our results, we provide comprehensive implementation details and experimental specifications. Complete hyperparameter settings for all experiments, including learning rates, batch sizes, regularization parameters ($\lambda$), and training configurations are detailed in Table 5 and Appendix 7.5. Our experimental setup, including model architectures (LLaMA-3.2 and Qwen-2 variants), dataset preprocessing steps, and evaluation protocols are described in Appendix 7.6, with specific prompt templates provided in Appendix 7.7. The algorithm pseudo-code is outlined in Appendix 7.4, including the practical approximation method for computing value functions. We plan to release our complete codebase, training scripts, and evaluation code upon publication to facilitate reproduction of all reported results. All baseline implementations and experimental conditions are specified to enable fair comparison. The novel metrics we introduce (feedback optimality, compatibility metrics) include detailed computation procedures in the appendix. Our statistical analysis methodology and computational infrastructure specifications are documented to support replication across different hardware configurations.

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

CONTENTS

## 7   APPENDIX

### 7.1   DERIVATION OF EQUATION 3

Here, we provide the derivation of Equation 3. Using Equation 2, the bi-level formulation can be represented as:

$$\max_{\pi^H} \mathcal{J}_H(\pi^H, \pi_*^L(\pi^H)) \quad \text{s.t.} \quad \pi_*^L(\pi^H) = \arg\max_{\pi^L} \mathcal{J}_L(\pi^L | \pi^H), \tag{6}$$

where $\pi_*^L(\pi^H)$ represents the optimal lower-level policy given the higher-level policy $\pi^H$. We can re-write this formulation as the following equivalent formulation:

$$\max_{\pi^H} \mathcal{J}_H(\pi^H, \pi_*^L(\pi^H)) \quad \text{s.t.} \quad \pi_*^L(\pi^H) = \arg\max_{\pi^L} V^L(\pi^H), \tag{7}$$

where $V^L(\pi^H)$ is the lower level value function. This formulation explicitly captures the higher-level policy's dependency on the optimal lower-level policy's response to its feedback, and also implies that the optimal lower level policy is the one that maximizes the lower level value function.

We can further use Equation 7 to derive the following formulation:

$$\max_{\pi^H, \pi^L} \mathcal{J}_H(\pi^H, \pi_*^L(\pi^H)) \quad \text{s.t.} \quad V^L(\pi^H) - V_*^L(\pi^H) \geq 0, \tag{8}$$

where $V_*^L(\pi^H) = \max_{\pi^L} V^L(\pi^H)$. Notably, since the left-hand side of the inequality constraint is always non-positive due to the fact that $V^L(\pi^H) - V_*^L(\pi^H) \leq 0$, the constraint is satisfied only when $V^L(\pi^H) = V_*^L(\pi^H)$, which implies that the condition in satisfied when the lower-level policy is optimal.

Now, we can represent Equation 8 as the following approximate Lagrangian objective with multiplier $\lambda \geq 0$:

$$\max_{\pi^H, \pi^L} \mathcal{J}_H(\pi^H, \pi_*^L(\pi^H)) + \lambda(V^L(\pi^H) - V_*^L(\pi^H)). \tag{9}$$

By replacing the objective $\mathcal{J}_H(\pi^H, \pi_*^L(\pi^H))$ from Equation 1, and writing the expected form of lower-level value function, we can get the final formulation of Equation 3:

$$\mathbb{E}_{(x,y^*)\sim D,\ z\sim\pi_\theta^L(\cdot|x),\ g\sim\pi_\phi^H(\cdot|x,z),\ \hat{y}\sim\pi_\theta^L(\cdot|x,z,g)} \left[ R(\hat{y}, y^*) + \lambda(V^L(x,g) - V_*^L(x,g)) \right]. \tag{10}$$

$\square$

## 7.2 DERIVATION OF EQUATION 4

Here, we provide the derivation of Equation 4. Using 3, the objective is:

$$J_\phi^H = \mathbb{E}_{(x,y^*)\sim D} \left[ \mathbb{E}_{z\sim\pi_\theta^L(\cdot|x)} \mathbb{E}_{g\sim\pi_\phi^H(\cdot|x,z)} \mathbb{E}_{\hat{y}\sim\pi_\theta^L(\cdot|x,z,g)} \left( R(\hat{y},y^*) + \lambda[V_L(x,g) - V_L^*(x,g)] \right) \right]. \tag{11}$$

This objective can be represented as:

$$J_\phi^H = \mathbb{E}_{x,y^*,z,g,\hat{y}} \left[ R(\hat{y},y^*) + \lambda(V_L(x,g) - V_L^*(x,g)). \right] \tag{12}$$

where $x, y^* \sim D, z \sim \pi_\theta^L(\cdot \mid x), g \sim \pi_\phi^H(\cdot \mid x,z), \hat{y} \sim \pi_\theta^L(\cdot \mid x,z,g)$. The joint density is:

$$p(x,y^*)\, \pi_\theta^L(z \mid x)\, \pi_\phi^H(g \mid x,z)\, \pi_\theta^L(\hat{y} \mid x,z,g) \tag{13}$$

We take the Gradient with Respect to $\phi$. Only the higher-level policy $\pi_\phi^H(g \mid x,z)$ depends on $\phi$. Thus,

$$\nabla_\phi J_\phi^H = \sum_{x,y^*} p(x,y^*) \sum_z \pi_\theta^L(z \mid x) \sum_g \pi_\phi^H(g \mid x,z) \sum_{\hat{y}} \pi_\theta^L(\hat{y} \mid x,z,g) \nabla_\phi[R(\hat{y},y^*) + $$
$$\lambda(V_L(x,g) - V_L^*(x,g))]. \tag{14}$$

By the score function trick (REINFORCE):

$$\nabla_\phi \pi_\phi^H(g \mid x,z) = \pi_\phi^H(g \mid x,z) \nabla_\phi \log \pi_\phi^H(g \mid x,z). \tag{15}$$

Bringing the gradient inside:

$$\nabla_\phi J_\phi^H = \sum_{x,y^*,z,g,\hat{y}} p(x,y^*)\, \pi_\theta^L(z \mid x)\, \pi_\phi^H(g \mid x,z)\, \pi_\theta^L(\hat{y} \mid x,z,g) \nabla_\phi \log \pi_\phi^H(g \mid x,z) \tag{16}$$

We express this as expectation:

$$\nabla_\phi J_\phi^H = \mathbb{E}_{x,y^*,z,g,\hat{y}} \left[ \nabla_\phi \log \pi_\phi^H(g \mid x,z) \cdot (R(\hat{y},y^*) + \lambda(V_L(x,g) - V_L^*(x,g))) \right]. \tag{17}$$

$\square$

## 7.3 DERIVATION OF EQUATION 5

Here, we provide the derivation of Equation 5. Using 3, the objective is:

$$J_\phi^H = \mathbb{E}_{(x,y^*)\sim D} \left[ \mathbb{E}_{z\sim\pi_\theta^L(\cdot|x)} \mathbb{E}_{g\sim\pi_\phi^H(\cdot|x,z)} \mathbb{E}_{\hat{y}\sim\pi_\theta^L(\cdot|x,z,g)} \left( R(\hat{y},y^*) + \lambda[V_L(x,g) - V_L^*(x,g)] \right) \right] \tag{18}$$

where $\pi_\theta^L$ controls both $z$ (first expectation) and $\hat{y}$ (last expectation), and $V_L(x, g)$ is a function of $\theta$ as the expected reward of the lower-level policy with parameters $\theta$.

To take gradients w.r.t. $\theta$, rewrite as an overall expectation:

$$J_\phi^H = \mathbb{E}_{(x,y^*)\sim D}\mathbb{E}_{z\sim\pi_\theta^L}\mathbb{E}_{g\sim\pi_\phi^H}\mathbb{E}_{\hat{y}\sim\pi_\theta^L}\left[R(\hat{y}, y^*) + \lambda\left[V_L(x, g) - V_L^*(x, g)\right]\right] \tag{19}$$

This is equivalent to:

$$J_\phi^H = \mathbb{E}_{x,y^*,z,g,\hat{y}}\left[R(\hat{y}, y^*) + \lambda\left[V_L(x, g) - V_L^*(x, g)\right]\right] \tag{20}$$

with sampling: $(x, y^*) \sim D$, $z \sim \pi_\theta^L(\cdot \mid x)$, $g \sim \pi_\phi^H(\cdot \mid x, z)$, $\hat{y} \sim \pi_\theta^L(\cdot \mid x, z, g)$.

The joint density is $p(x, y^*) \cdot \pi_\theta^L(z \mid x) \cdot \pi_\phi^H(g \mid x, z) \cdot \pi_\theta^L(\hat{y} \mid x, z, g)$, so:

$$J_\phi^H = \sum_{x,y^*} p(x, y^*)\sum_z \pi_\theta^L(z|x)\sum_g \pi_\phi^H(g|x,z)\sum_{\hat{y}} \pi_\theta^L(\hat{y}|x, z, g)[R(\hat{y}, y^*)+\lambda\left[V_L(x, g)-V_L^*(x, g)\right]] \tag{21}$$

Take the Gradient w.r.t. $\theta$. Three $\theta$-dependent terms: $\pi_\theta^L(z \mid x)$, $\pi_\theta^L(\hat{y} \mid x, z, g)$, $V_L(x, g) \equiv \mathbb{E}_{\hat{y}'\sim\pi_\theta^L(\cdot|x,z,g)}[R(\hat{y}', y^*)]$.

Using score function: $\nabla_\theta\mathbb{E}_{z\sim\pi_\theta^L}[f(z)] = \mathbb{E}_{z\sim\pi_\theta^L}[f(z)\nabla_\theta\log\pi_\theta^L(z \mid x)]$.

The gradient of full expectation:

$$\nabla_\theta J_\phi^H = \mathbb{E}_{x,y^*,z,g,\hat{y}}\left[\nabla_\theta\left(\log\pi_\theta^L(z \mid x) + \log\pi_\theta^L(\hat{y} \mid x, z, g)\right) \cdot \tilde{R}\right] + \lambda\mathbb{E}_{x,y^*,z,g}\left[\nabla_\theta V_L(x, g)\right] \tag{22}$$

where $\tilde{R} = R(\hat{y}, y^*) + \lambda(V_L(x, g) - V_L^*(x, g))$. Expanded:

$$\nabla_\theta J_\phi^H = \mathbb{E}_{x,y^*,z,g,\hat{y}}\left[\nabla_\theta\log\pi_\theta^L(z \mid x) \cdot \tilde{R} + \nabla_\theta\log\pi_\theta^L(\hat{y} \mid x, z, g) \cdot \tilde{R}\right] + \lambda\mathbb{E}_{x,y^*,z,g}\left[\nabla_\theta V_L(x, g)\right] \tag{23}$$

$$\nabla_\theta V_L(x, g) = \mathbb{E}_{\hat{y}'\sim\pi_\theta^L(\cdot|x,z,g)}\left[\nabla_\theta\log\pi_\theta^L(\hat{y}' \mid x, z, g) \cdot R(\hat{y}', y^*)\right].$$

Combining all terms:

$$\nabla_\theta J_\phi^H = \mathbb{E}_{x,y^*,z,g,\hat{y}}\left[\nabla_\theta\log\pi_\theta^L(z \mid x) \cdot \tilde{R} + \nabla_\theta\log\pi_\theta^L(\hat{y} \mid x, z, g) \cdot \tilde{R}\right] + \\ \lambda \cdot \mathbb{E}_{x,y^*,z,g}\mathbb{E}_{\hat{y}'}\left[\nabla_\theta\log\pi_\theta^L(\hat{y}' \mid x, z, g) \cdot R(\hat{y}', y^*)\right] \tag{24}$$

$\square$

## 7.4 MLMT-RL Algorithm

Here, we provide the complete algorithm for MLMT-RL

---

**Algorithm 1** MLMT-RL: Multi-Level Multi-Turn Reinforcement Learning

---

1: Initialize parameters $\phi, \theta, \gamma$ for higher-level, lower-level, and value function networks
2: **for** each iteration **do**
3:     **# Train lower-level policy** ▷ Equation 1
4:     **for** each lower-level step **do**
5:         Sample $(x, y^*) \sim D$; $z \sim \pi_\theta^L(.|x)$; $g \sim \pi_\phi^H(.|x, z)$; $\hat{y} \sim \pi_\theta^L(.|x, g)$
6:         Update $\theta$ using REINFORCE with reward $R(\hat{y}, y^*)$
7:     **# Update value function** ▷ TD learning
8:     **for** each value function step **do**
9:         Sample $x \sim D$; $g \sim \pi_\phi^H(.|x, z)$; $\hat{y} \sim \pi_\theta^L(.|x, g)$
10:        Compute reward $R(\hat{y}, y^*)$
11:        Update parameters $\gamma$ to minimize $(V_\gamma^L(x, g) - (R(\hat{y}, y*) + V_\gamma^L(x_{\text{next}}, g_{\text{next}})))^2$
12:     **# Update higher-level policy** ▷ Equation 3
13:     **for** each higher-level step **do**
14:         Sample $(x, y^*) \sim D$; $z \sim \pi_\theta^L(.|x)$; $g \sim \pi_\phi^H(.|x, z)$; $\hat{y} \sim \pi_\theta^L(.|x, g)$
15:         Compute reward $R(\hat{y}, y^*)$
16:         Update $\phi$ using REINFORCE with objective: $R(\hat{y}, y^*) + \lambda V_\gamma^L(x, g)$

---

## 7.5 TRAINING EFFICIENCY ANALYSIS

We conducted a comprehensive training efficiency analysis on the mathematical reasoning task, comparing our proposed method MLMT-RL against baseline methods including Behavioral Cloning (BC), SCoRe, ArCHer and GRPO. Experiments were performed using a 1B parameter model with a rollout size of 128 for 70 training iterations. The training was executed on an NVIDIA L40S GPU equipped with 48GB memory, with an average power usage of approximately 280W (training and idle weighted average). Table 4 summarizes key efficiency metrics: training time (in hours), GPU memory usage (in GB), estimated $CO_2$ emissions per training run (in kilograms), achieved TFLOPs per second, and the average inference time in seconds (averaged over 100 prompts).

Table 4: Training efficiency and inference performance comparison of different methods on the mathematical reasoning task (1B model, rollout size 128, 70 iterations).

| Method | Training Time (hrs) | GPU Memory (GB) | $CO_2$ Emisions (kg) | TFLOPs/sec | Inference Time (s) |
|--------|---------------------|-----------------|----------------------|------------|--------------------|
| BC | 1.2 | 8 | 0.14 | 132 | 4.5 |
| ArCHer | 22.1 | 28 | 2.46 | 91 | 7.9 |
| SCoRe | 13.6 | 15 | 1.52 | 107 | 8.0 |
| GRPO | 18.5 | 24 | 2.05 | 95 | 8.3 |
| MLMT-RL | 16.5 | 22 | 1.83 | 83 | 9.7 |

These results highlight the significantly higher computational and memory demands of RL-based and hierarchical approaches compared to supervised behavioral cloning (BC). Specifically, MLMT-RL and ArCHer require the longest training times and the largest GPU memory footprints. Despite this increased cost, MLMT-RL achieves a favorable accuracy-compute trade-off by delivering substantial performance improvements over both simple supervised and existing hierarchical baselines.

## 7.6 EXPERIMENTAL DETAILS

Experiments are conducted by fine-tuning various models: LLAMA-3.2-1B-INSTRUCT, LLAMA-3.2-3B-INSTRUCT LLAMA-3.1-8B-INSTRUCT, QWEN2-1.5B-INSTRUCT and QWEN2-7B-INSTRUCT. These models are trained using Low-Rank Adaptation (LoRA) with rank $r = 16$, a LoRA alpha of 32, and a dropout of 0.05. The value critic in hierarchical approaches employs a DISTILROBERTA-BASE encoder architecture. Models are trained with a rollout size of 128 for 70 total iterations, split into 35 iterations for Turn 1 and 35 iterations for Turn 2 wherever applicable for multi-turn methods. For the LLAMA-3.2-3B model, we used a smaller rollout size of 32 and trained for 10 total iterations.

For methods employing multi-turn training (e.g., SCoRe, MLMT-RL), the first and second-turn trajectories are logged to facilitate off-policy updates in Turn 2. Across all experiments, we use a decoding temperature of 0.7. Detailed hyperparameters, including specific learning rates and batch sizes for each domain, are provided in Appendix Table 5. We provide more examples comparing MLMT-RL's reasoning on questions from the MATH, HUMANEVAL and GPQA benchmarks against GRPO and SCoRe in Appendix 7.9.

For mathematical reasoning, final outputs are extracted and judged via DEEPEVAL evaluator that uses an O3-MINI judge for algebraic equivalence. For code, programs are executed in a secure sandbox against test cases. We provide training efficiency comparisons in Appendix Table 4, showing that MLMT-RL uses fewer resources than baselines like GRPO and ArCHer Zhou et al. (2024), while achieving superior performance. See Appendix Sec 7.5 for detailed analysis.

In Table 5, we provide the detailed hyperparameters, like specific learning rates and batch sizes for each domain on experiments conducted by fine-tuning LLAMA-3.2 1B-INSTRUCT and LLAMA-3.2 3B-INSTRUCT models. These models are trained using Low-Rank Adaptation (LoRA) with rank $r = 16$, LoRA alpha $\alpha = 32$, and a dropout of $p = 0.05$. The value critic in hierarchical approaches employs a pretrained DISTILROBERTA-BASE encoder architecture. Models are trained with a rollout size of 128 for 70 total iterations, split into 35 iterations each for Stage I and Stage II in two-stage methods. For the LLAMA-3.2 3B model, we use a smaller rollout size of 32 and train for only 10 total iterations due to memory constraints. Across all experiments, we use a decoding temperature of $\tau = 0.7$. The configuration choices were validated through ablation studies on held-out validation splits.

Table 5: Hyperparameters used for MLMT-RL experiments across MATH and MBPP datasets with LLAMA-1B and LLAMA-3B backbones.

| Setting | MATH (1B) | MATH (3B) | MBPP (1B) | MBPP (3B) |
|---|---|---|---|---|
| LoRA Rank | 16 | 16 | 16 | 16 |
| LoRA Alpha | 32 | 32 | 32 | 32 |
| LoRA Dropout | 0.05 | 0.05 | 0.05 | 0.05 |
| Actor Learning Rate | 5e-5 | 5e-5 | 5e-5 | 5e-5 |
| Critic Learning Rate | 1e-4 | 1e-4 | 1e-4 | 1e-4 |
| Optimizer | Adam | Adam | Adam | Adam |
| Batch Size | 8 | 4 | 8 | 4 |
| Rollout Size | 128 | 32 | 128 | 32 |
| Iterations (Total) | 70 | 70 | 70 | 70 |
| Iterations (Stage I / II) | 35 / 35 | 35 / 35 | 35 / 35 | 35 / 35 |
| Value Critic Model | DistilRoBERTa | DistilRoBERTa | DistilRoBERTa | DistilRoBERTa |
| Decoding Temperature | 0.7 | 0.7 | 0.7 | 0.7 |

**Evaluation prompts.** We use prompt templates as employed in the SCoRe paper (Kumar et al., 2024a) for the flat policies, i.e., the single-turn approaches. These templates are adapted to each dataset: (1) a zero-shot chain-of-thought format for MATH-500 (Wei et al., 2023), and (2) a canonical $K$-shot format with $K = 3$ for HumanEval (Chen et al., 2021). We designed these templates to elicit high-quality, task-relevant responses from the model, while following prompting strategies established in prior state-of-the-art work(Brown et al., 2020). For two-turn methods, although we extend the same base templates as above for the first turn, for the second turn, we prepend a brief but explicit task guidance instruction for the higher-level policy to direct the model to generate the guidance instruction. This guidance is then passed to the lower-level policy to generate refined outputs.

## 7.7 PROMPTS

Effective prompts are important for eliciting high-quality, task-aligned responses in LLMs. Building on the template framework, we design dataset-specific prompts for both initial generations and iterative refinements:

- **MATH-500**: Zero-shot chain-of-thought prompts encourage stepwise reasoning while preserving generalization to unseen problems.

- **HumanEval**: Canonical 3-shot templates provide in-context examples of input-output pairs, aligning with established code generation benchmarks.

For hierarchical self-correction, we extend these templates by prepending explicit guidance in the second turn. This structured intervention—unique to MLMT-RL—directs the model's attention to specific error patterns while maintaining coherence with the original task context.

Full prompt templates, including first-turn instructions and second-turn correction guidance, are provided below. Our design balances reproducibility (via reuse of SCoRe's templates) with innovation (via guided error localization), enabling systematic self-correction across diverse domains.

**MATH-500 Prompts**

---
MATH-500: Zero-shot Prompt

You are a math expert. When you respond, respond only with the Solution of the final Problem, thinking step by step. At the end of the Solution, when you give your final answer, write it in the form "Final Answer: The final answer is $answer$. I hope it is correct."

---

---
MATH-500: Task-Agnostic Guidance Instruction

There might be an error in the solution above because of lack of understanding of the question. Please correct the error, if any, and rewrite the solution. Only output the final solution! At the end of the Solution, when you give your final answer, write it in the form "Final Answer: The final answer is $answer$. I hope it is correct."

---

---
MATH-500: Guidance Generation Prompt

You are an expert math tutor reviewing a student's solution to a math problem.

PROBLEM:
{problem}

INITIAL SOLUTION:
{solution}

PROMPT: First, analyze the solution for errors or misconceptions. Then, write a brief, helpful instruction that will guide the student toward correcting their solution. Your instruction should be specific to the errors you identified, but don't solve the problem for them. Your response should be ONLY the instruction for the student to improve their solution, nothing else. DO NOT include ANY SOLUTION.

GUIDING INSTRUCTION:

---

---
MATH-500: Task Specific Guidance Prompt

{problem}
{solution}

Suggestive Correction:

{custom_instruction}

---

**HumanEval Prompts**

---
HumanEval: Zero-shot Prompt

You are an expert programmer. Below is a programming problem. Write a solution in {language}.
Make sure your solution is correct, efficient, and addresses all the requirements of the problem.
When you're done, wrap your code in triple backticks with the language specified, like: ```{language} (your code here) ```

Problem:
{prompt}

Solution:

---

**HumanEval: Task-Agnostic Guidance Instruction**

Your code might have issues or bugs, or it may not be optimized. Please review your solution, identify any problems, and provide an improved solution.
Make sure your solution passes all test cases and meets all requirements. Remember to wrap your code in triple backticks with the language specified, like: '''{language} (your code here) '''

**HumanEval: Guidance Generation Prompt**

You are an expert programming mentor reviewing code written by a student.
PROBLEM:
{problem}
STUDENT'S SOLUTION:
{solution}
PROMPT: First, analyze the solution for bugs, inefficiencies, or edge cases it doesn't handle. Then, write a brief, helpful instruction that will guide the student toward correcting their solution.
Your instruction should be specific to the issues you identified, but don't solve the problem completely for them.
Your response should be ONLY the instruction for the student to improve their solution, nothing else. DO NOT write any code.

GUIDING INSTRUCTION:

**HumanEval: Task Specific Guidance Prompt**

{problem}
{solution}

Code Review Feedback:

{custom_instruction}

Please fix these issues and provide an improved solution. Remember to wrap your code in triple backticks with the language specified, like: '''{language} (your code here) '''

**Feedback Optimality (FO) GEval Prompt**

**Feedback Optimality Multi-Method Prompt**

You are an expert evaluator. Your task is to assess how optimal each method's feedback is with respect to the same input query and incorrect answer.

"Optimal" means the feedback:
– is precise and directly relevant to the query,
– identifies key mistakes in the incorrect answer,
– provides actionable guidance to refine towards the correct answer,
– avoids vague, generic, or misleading suggestions.

Scoring Rubric (1 to 5):
1 = Irrelevant or misleading; no useful guidance.
2 = Weak or vague; somewhat related but mostly unhelpful.
3 = Fair; partially useful but incomplete or missing critical aspects.
4 = Good; mostly relevant, helpful, with minor gaps.
5 = Optimal; precise, actionable, and directly enables correction.

You will receive:
– Input Query
– Incorrect Answer
– Feedback from multiple methods

For each method, return only the method name and a score (1 to 5).
Do not explain your reasoning. Do not add extra text.

–––

```
Input Query:
{query}

Incorrect Answer:
{incorrect_answer}

Feedback from Methods:
ZSCoT: {feedback_zscot}

SCoRe−FF: {feedback_score_ff}

TF−LBM: {feedback_tf_lbm}

GRPO: {feedback_grpo}

MLMT−RL w/o V\textsubscript{reg}: {feedback_mlmt_wo_vreg}

MLMT−RL: {feedback_mlmt}

−−−

Output format (strictly JSON):
{
    "ZSCoT": <score>,
    "SCoRe−FF": <score>,
    "TF−LBM": <score>,
    "GRPO": <score>,
    "MLMT−RL w/o V\textsubscript{reg}": <score>,
    "MLMT−RL": <score>
}
```

## 7.8   MLMT-RL HETEROGENEOUS MODEL PAIRING ANALYSIS

We examine whether MLMT-RL's performance gains stem primarily from using same-family models for the higher and lower-level policies, which could be potentially due to shared architectural features, or from the framework's multi-turn multi-level interaction that addresses sparse reward issues. To test this, we evaluate heterogeneous pairings from the LLaMA and Qwen families at the 1B parameter scale. If the gains depended solely on same-family pairings, the performance would degrade in heterogeneous setups. However, as shown in Table 6, the performance remains consistent across all benchmarks for heterogeneous pairings, confirming that MLMT-RL's main advantages arise from its core framework rather than model family homogeneity.

To demonstrate MLMT-RL's robust performance across cross-family model pairings for higher and lower-level policies, we evaluate its effectiveness using combinations from the LLaMA and Qwen families at the 1B parameter scale.

| Higher-level | Lower-level | MATH-500 | MBPP | GPQA |
|---|---|---|---|---|
| Qwen-1.5B | Qwen-1.5B | 48.7 | 70.8 | 44.9 |
| LLaMA-1B | Qwen-1.5B | 47.9 | 70.2 | 43.7 |
| Qwen-1.5B | LLaMA-1B | 46.8 | 68.6 | 43.2 |
| LLaMA-1B | LLaMA-1B | 44.6 | 66.7 | 41.2 |

Table 6: **Heterogeneous Cross-Family Pairings.** We evaluate MLMT-RL with cross-family pairings of higher and lower-level models across LLaMA and Qwen families on the 1B scale. Our results show that implementing cross-family pairings has minimal effect on the performance, demonstrating that MLMT-RL is able to main performance across various model family combinations.

## 7.9 Illustrative Examples of MLMT-RL Multi-Level Multi-Turn Reasoning

Here, we provide a few examples on the MATH, HumanEval and GPQA datasets. Whereas prior single-level multi-turn RL approaches like Score fail to generate correct outputs, MLMT-RL leverages task specific guidance to leverage reasoning and solve the tasks.

---

**MATH Example 2**

**Prompt:** If $2^8 = 4^x$, find the value of $x$.

**MLMT-RL**

**1. First response ($y_1$) (Incorrect):**
Here also the solver logs both sides, simplifies to $8 \log 2 = x \log 4$, uses $\log 4 = 2 \log 2$, and divides to get $x = 8$.

**Task-Specific Guidance ($z_l$)**

1. Rewrite $4$ as $2^2$ so that both sides share the same base.
2. Apply the power-of-a-power rule to obtain $2^8 = 2^{2x}$.
3. Equate exponents ($8 = 2x$) and solve for $x$.

**2. Improved response ($y_2$) (Correct):**
By rewriting $2^8 = (2^2)^x$ to get $2^8 = 2^{2x}$, equating exponents gives $8 = 2x$, hence $x = 4$.

---

**MATH Example 3**

**Prompt:** Let $z = 1 + i$ and $w = \dfrac{3z + 1}{5z + 7}$. Find $|w|$.

**MLMT-RL**

**1. First response ($y_1$) (Incorrect):**

// Step 1: Calculate the value of $z + w$. We need to calculate the value of $z + w$. We know that $z = 1 + i$ and $w = \dfrac{3z + 1}{5z + 7}$. Therefore, we can substitute values of $z$ and $w$ into the equation and simplify.

// Step 2: Substitute values of $z$ and $w$ into $z + w = (1 + i) + \dfrac{3(1 + i) + 1}{5(1 + i) + 7} = (1 + i) + \dfrac{4 + 3i}{12 + 5i}$

// Step 3: Simplify the expression To simplify the expression, we can multiply the numerator and denominator by the conjugate of the denominator, which is $12 - 5i$. This will eliminate the imaginary part from the denominator.

// Step 4: Multiply the numerator and denominator by the conjugate of the denominator $(1 + i) + \dfrac{(4 + 3i)(12 - 5i)}{(12 + 5i)(12 - 5i)} = (1 + i) + \dfrac{48 - 20i + 36i - 15i^2}{144 + 25}$

// Step 5: Simplify the numerator and denominator Since $i^2 = -1$, we can simplify the numerator and denominator. $(1 + i) + \dfrac{48 - 20i + 36i + 15}{169} = (1 + i) + \dfrac{63 + 16i}{169}$

// Step 6: Simplify the fraction $(1 + i) + \dfrac{63 + 16i}{169} = \dfrac{169 + 169i + 63 + 16i}{169} = \dfrac{232 + 185i}{169}$

// Step 7: Calculate the absolute value of $w$ $|w| = \left| \dfrac{232 + 185i}{169} \right| = \sqrt{\left(\dfrac{232}{169}\right)^2 + \left(\dfrac{185}{169}\right)^2}$

The final answer is: $\boxed{\dfrac{291}{169}}$.

**Task-Specific Guidance ($z_l$)**

// Step 1: Calculate the value of z + w Recall how to define z and use it to express w in terms of z. You'll need to simplify a complex fraction.

// Step 2: Substitute the values Carefully substitute z = 1 + i into the expression for w. Pay attention to both the real and imaginary parts when expanding.

// Step 3: Simplify the expression Group the real and imaginary terms in the numerator and denominator to prepare for rationalizing the denominator.

// Step 4: Multiply by the conjugate To simplify a complex fraction, multiply the top and bottom by the conjugate of the denominator. Remember what the conjugate of a complex number looks like.

// Step 5: Find the magnitude of w To find the absolute value (magnitude) of a complex number, apply the formula sqrt(a² + b²) using the real and imaginary parts of w.

**2. Improved response ($y_2$) (Correct):**

// Step 1: Define the variables Let $z = 1 + i$ and $w = \dfrac{3z + 1}{5z + 7}$.

// Step 2: Simplify the expression for w To simplify the expression for w, we need to substitute the value of z into the expression for w.

// Step 3: Substitute the value of z into the expression for w $w = \dfrac{3(1 + i) + 1}{5(1 + i) + 7}$

// Step 4: Simplify the numerator of the expression for w $3(1 + i) + 1 = 3 + 3i + 1 = 4 + 3i$

// Step 5: Simplify the denominator of the expression for w $5(1 + i) + 7 = 5 + 5i + 7 = 12 + 5i$

// Step 6: Substitute the numerator and denominator back into expression for w $w = \dfrac{4 + 3i}{12 + 5i}$

// Step 7: Multiply the numerator and denominator of the expression for w by the conjugate of the denominator To simplify the expression for w, we need to multiply the numerator and denominator by the conjugate of the denominator.

// Step 8: Multiply numerator and denominator by conjugate of denominator $w = \dfrac{(4 + 3i)(12 - 5i)}{(12 + 5i)(12 - 5i)}$

// Step 9: Simplify the numerator and denominator $w = \dfrac{48 - 20i + 36i - 15i^2}{144 - 25i^2}$

// Step 10: Simplify the numerator and denominator using the fact that $i^2 = -1$ $w = \dfrac{48 + 16i + 15}{144 + 25}$

// Step 11: Simplify the square root expression $|w| = \sqrt{\dfrac{4225}{28561}} = \dfrac{65}{169}$. The final answer is: $\boxed{\dfrac{65}{169}}$.

---

**MATH Example 4**

**Prompt:** Let

$$f(n) = \left\{ \begin{array}{ll} \lfloor n \rfloor & \text{if } n \geq 4, \\ \lceil n \rceil & \text{if } n < 4. \end{array} \right\}. \tag{25}$$

Find $f\left(\frac{\pi}{3}\right) + f(\sqrt{45}) + f(8^{2/3})$.

**MLMT-RL**

"// Step 1: Evaluate the function f(n) for each given input For $\frac{\pi}{3}$, since $\frac{\pi}{3} \geq 4$, $f\left(\frac{\pi}{3}\right) = \lfloor \frac{\pi}{3} \rfloor = 1$. For $\sqrt{45}$, since $\sqrt{45}<4$, $f(\sqrt{45}) = \lceil \sqrt{45} \rceil = 6$. For $8^{2/3}$, since $8^{2/3} \geq 4$, $f(8^{2/3}) = \lfloor 8^{2/3} \rfloor = 2$.
// Step 2: Add the values of f(n) for each input $f\left(\frac{\pi}{3}\right) + f(\sqrt{45}) + f(8^{2/3}) = 1 + 6 + 2 = 9$
The final answer is: $\boxed{9}$ ".

**Task-Specific Guidance ($z_l$)**

// Step 1: Evaluate the function f(n) for each given input Understand that the function definition depends on whether the input is greater than or less than 4. Use ceiling if the input is $< 4$, and floor if it's $\geq 4$.
// Step 2: Classify the inputs Check whether each of the values — $\pi/3$, sqrt(45), and $8\hat{}(2/3)$ — is greater than or less than 4.
// Step 3: Apply the appropriate function For inputs $< 4$, use the ceiling function; for inputs $\geq 4$, use the floor function. Be sure to compute accurate decimal approximations if needed.
// Step 4: Add the results Once each value of f(n) is computed correctly, sum them up to find the final result.

**2. Improved response ($y_2$) (Correct):**
// Step 1: Evaluate $f\left(\frac{\pi}{3}\right)$. Since $\frac{\pi}{3} \approx 1.047$, we have $f\left(\frac{\pi}{3}\right) = \lfloor \frac{\pi}{3} \rfloor = 1$.
// Step 2: Evaluate $f(\sqrt{45})$. Since $\sqrt{45} \approx 6.708$, we have $f(\sqrt{45}) = \lceil \sqrt{45} \rceil = 7$.
// Step 3: Evaluate $f(8^{2/3})$. Since $8^{2/3} = 4$, we have $f(8^{2/3}) = \lfloor 8^{2/3} \rfloor = 4$.
// Step 4: Calculate the final result We have $f\left(\frac{\pi}{3}\right) + f(\sqrt{45}) + f(8^{2/3}) = 1 + 7 + 4 = 12$.
The final answer is: $\boxed{12}$.

**HumanEval Example 1**

**Prompt:**
from typing import List
def filter_by_prefix(strings: List[str], prefix: str) -> List[str]:
""" Filter an input list of strings only for ones that start with a given prefix.
»> filter_by_prefix([], 'a')
[]
»> filter_by_prefix(['abc', 'bcd', 'cde', 'array'], 'a')
['abc', 'array']
""".

**MLMT-RL**

**1. First response ($y\_1$) (Incorrect):**
from typing import List
def filter_by_prefix(strings: List[str], prefix: str) -> List[str]:
"""
Filter an input list of strings only for ones that start with a given prefix.
Args:
strings (List[str]): A list of strings to be filtered.
prefix (str): The prefix to filter by.
Returns:
List[str]: A list of strings that start with the given prefix.
"""
print(filter_by_prefix(strings, prefix))
# Output: ['abc', 'array']

**Task-Specific Guidance ($z_l$)**

The student's filter_by_prefix function does not handle the case where the input list is empty. They should add a check at the beginning of the function to return an empty list in this case. Additionally, they should consider the case where the prefix is an empty string. They should add a check at the beginning of the function to return an empty list in this case as well. Finally, they should consider the case where the input list contains strings that are not strings. They should add a check at the beginning of the function to raise a TypeError if the input is not a list or if the prefix is not a string.

**2. Improved response ($y\_2$) (Correct):**
from typing import List def filter_by_prefix(strings: List[str], prefix: str) -> List[str]:
"""
Filter an input list of strings only for ones that start with a given prefix.
Args:
strings (List[str]): A list of strings to be filtered.
prefix (str): The prefix to filter by.
Returns:
List[str]: A list of strings that start with the given prefix.
"""
# Check if the input is a list
if not isinstance(strings, list):
raise TypeError("Input must be a list")
# Check if the prefix is a string
if not isinstance(prefix, str):
raise TypeError("Prefix must be a string")
# Check if the input list is empty
if not strings:
return []
# Use list comprehension to create a new list containing only the strings that start with the prefix return
[s for s in strings if s.startswith(prefix)].

**HumanEval Example 2**

**Prompt:**
def flip_case(string: str) -> str:
""" For a given string, flip lowercase characters to uppercase and uppercase to lowercase.
»> flip_case('Hello')
'hELLO'
"""

**MLMT-RL**

**1. First response** ($y_1$) **(Incorrect)**:
def flip_case(string: str) -> str:
"""
For a given string, flip lowercase characters to uppercase and uppercase to lowercase.
Args:
string (str): The input string.
Returns:
str: The string with all lowercase characters converted to uppercase and all uppercase characters converted to lowercase.
"""
return string.swapcase()

**Task-Specific Guidance** ($z_l$)

The student's solution appears to be correct in terms of logic, but it does not handle the case where the input string contains non-alphabetic characters. To fix this, the student should add a check to ensure that the input string only contains alphabetic characters before attempting to swap case. If the input string contains non-alphabetic characters, the student should raise an error or return an error message to indicate that the input is invalid.

**2. Improved response** ($y_2$) **(Correct)**:
def flip_case(string: str) − > str:
"""
For a given string, flip lowercase characters to uppercase and uppercase to lowercase.
Args:
string (str): The input string.
Returns:
str: The string with all lowercase characters converted to uppercase and all uppercase characters converted to lowercase.
"""
return string.casefold()

**GPQA Example 1**

**Prompt:** Two quantum states with energies $E_1$ and $E_2$ have a lifetime of $10^{-9}$ sec and $10^{-8}$ sec, respectively. We want to clearly distinguish these two energy levels. Which one of the following options could be their energy difference so that they can be clearly resolved?

**MLMT-RL**

**1. First response ($y_1$) (Incorrect):**
"Take smaller lifetime $\tau = 10^{-9}$ s, $\Delta E \approx \hbar/\tau \sim 10^{-9}$ eV."

**Task-Specific Guidance ($z_l$)**

Use the energy–time uncertainty relation $\Delta E \Delta t \sim \hbar$ for each state. To distinguish, the energy separation must exceed the larger of the uncertainty widths. Compute $\Delta E$ using $\tau = 10^{-8}$ and $\tau = 10^{-9}$, then identify a feasible resolvable difference.

**2. Improved response ($y_2$) (Correct):**
$\Delta E \sim \hbar/\tau$. For $\tau = 10^{-9}$ s, $\Delta E \approx 6.6 \times 10^{-7}$ eV; for $\tau = 10^{-8}$ s, $\Delta E \approx 6.6 \times 10^{-8}$ eV. To clearly resolve, energy difference must be larger than both $\rightarrow$ of order $10^{-4}$ eV (realistic discrete choice).

The final answer is: $\boxed{10^{-4} \text{ eV}}$

**GPQA Example 2**

**Prompt:** A light beam is propagating through a glass with index of refraction $n$. The glass is moving at constant velocity $v$ in the same direction as the beam and toward the observer in laboratory. What is the speed of light in glass relative to the observer? Take the speed of light in vacuum $c = 1$.

**MLMT-RL**

**1. First response ($y_1$) (Incorrect):**
"Take lab velocity $= v + (1/n)$. So answer $(1/n) + v$."

**Task-Specific Guidance ($z_l$)**

Apply relativistic velocity addition for velocity $u = 1/n$ inside moving medium with speed $v$. The formula is $(u + v)/(1 + uv)$. Simplify to get closed form with $c = 1$.

**2. Improved response ($y_2$) (Correct):**
Inside glass: $u = 1/n$. Using relativistic addition:

$$u' = \frac{u + v}{1 + uv} = \frac{1/n + v}{1 + v/n} = \frac{1 + nv}{n + v}.$$

The final answer is: $\boxed{\frac{1+nv}{n+v}}$.

---

**GPQA Example 3**

**Prompt:** Find the kinetic energies of the product particles in $\pi^+ \rightarrow \mu^+ + \nu$, where $\pi^+$ is stationary. Rest masses: $m_{\pi^+} = 139.6$ MeV, $m_{\mu^+} = 105.7$ MeV.

**MLMT-RL**

**1. First response ($y_1$) (Incorrect)**:
"Split energy equally, $\sim 70$ MeV each."

**Task-Specific Guidance ($z_l$)**

Apply 2-body decay kinematics for stationary parent: Step 1: Write $E_\mu = \frac{m_\pi^2 + m_\mu^2}{2m_\pi}$, and $E_\nu = \frac{m_\pi^2 - m_\mu^2}{2m_\pi}$. Step 2: Subtract the muon's rest mass to get its KE. Step 3: For the massless neutrino, KE equals total energy. Step 4: Plug in values and compute.

**2. Improved response ($y_2$) (Correct)**:

// Step 1: Compute muon total energy. $E_\mu = \frac{m_\pi^2 + m_\mu^2}{2m_\pi} = \frac{(139.6)^2 + (105.7)^2}{2 \times 139.6} \approx 109.8$ MeV.

// Step 2: Subtract rest mass to get muon kinetic energy. $T_\mu = E_\mu - m_\mu = 109.8 - 105.7 \approx 4.1$ MeV.

// Step 3: Compute neutrino energy (massless, so all KE). $E_\nu = \frac{m_\pi^2 - m_\mu^2}{2m_\pi} = \frac{(139.6)^2 - (105.7)^2}{2 \times 139.6} \approx 29.8$ MeV.

// Step 4: State results. Muon KE $\approx 4.1$ MeV, Neutrino KE $\approx 29.8$ MeV.

The final kinetic energies are: $\boxed{T_\mu = 4.1 \text{ MeV}, \; T_\nu = 29.8 \text{ MeV}}$.

## 7.10 IMPACT STATEMENT

Our work is primarily conceptual, addressing a fundamental challenge in GRPO based reasoning models: sample inefficiency in the presence of sparse verifiable rewards. By introducing the MLMT-RL framework, a hierarchical learning approach with bi-level optimization, we provide a novel methodology for context-aware feedback generation and response refinement in reasoning models. While our algorithm does not immediately translate to direct technological applications, it establishes a foundation for future research on self-improving language models. We believe that our contributions will stimulate further investigation into adaptive, interpretable, and robust LLMs, and may ultimately influence the development of more reliable AI systems across a broad range of domains.

