# OpenReview forum: "Multi-Level Multi-Turn RL Outperforms GRPO: Reasoning with Textual Feedback"
_ICLR.cc/2026/Conference — Submitted to ICLR 2026_

### Official Review · Reviewer_wXhB · 2025-10-15

**Soundness:** 2
**Presentation:** 3
**Contribution:** 3
**Rating:** 6
**Confidence:** 3

**Summary:**

This paper proposes Multi-Level Multi-Turn Reinforcement Learning (MLMT-RL), a novel hierarchical reinforcement learning framework for reasoning tasks. The core idea is to mitigate the sparse reward problem in GRPO (Group Relative Policy Optimization) by using a higher-level policy to generate task-specific textual feedback, which guides a lower-level policy to refine responses. The method features:
A bi-level optimization structure with value regularization. New evaluation metrics (feedback optimality and compatibility). Significant empirical gains over GRPO and other strong baselines (e.g., SCoRe, ArCHer).

**Strengths:**

The paper addresses the sparse reward problem in GRPO-style RL by introducing a multi-level, multi-turn framework that leverages textual feedback for improved learning efficiency.

The bi-level optimization formulation with value function regularization is reasonable and aligns the feedback generation with the refinement process.

Empirical results show solid gains across multiple reasoning benchmarks, with smaller models outperforming larger GRPO baselines.

Novel metrics for feedback quality and compatibility provide a useful way to evaluate intermediate learning signals

**Weaknesses:**

The proposed MLMT-RL framework presents a reasonable yet mostly compositional innovation. While the idea of multi-turn feedback-based refinement has been explored in various prior works, this paper organizes it into a bi-level reinforcement learning framework with value function regularization, and introduces evaluation metrics to quantify feedback quality. The contribution is more in the systematization and empirical demonstration than in the introduction of novel algorithmic paradigms.

The overall architecture is a fairly straightforward composition of known ideas (multi-turn feedback, hierarchical RL), with limited algorithmic novelty.

The textual feedback mechanism relies heavily on LLM heuristics, and the feedback generation process is not end-to-end learnable.

Experiments are restricted to structured reasoning tasks (math/code/QA), and generalization to open-ended or real-world scenarios is unclear.

**Questions:**

Generalization / Task Scope

Have the authors considered applying the method to open-ended or real-world tasks beyond structured reasoning benchmarks?

Robustness of Feedback

How robust is the approach to noisy or hallucinated feedback? Would incorrect guidance from the higher-level model degrade performance?

Ablation / Design Choice

Is there any analysis on how much the value regularization term contributes to the performance, compared to standard two-level training?

Model Efficiency

Would the method still hold advantages in settings where training two separate models is computationally prohibitive?

---

> ### Author Response · Authors · 2025-11-26
> **Response to Reviewer wXhB [Part 1]**
>
> We would like to express our gratitude to the reviewer for dedicating their valuable time and effort towards evaluating our manuscript, which has allowed us to strengthen the manuscript. We deeply appreciate the insightful feedback provided, and we have thoroughly responded to reviewer's inquiries in the responses provided below.
>
> ### Response to Weaknesses
>
> > **W1:** The proposed MLMT-RL framework presents a reasonable ... with limited algorithmic novelty.
>
> **Response:** We respectfully disagree with the characterization of MLMT-RL as straightforward composition. While we build on established concepts in multi-turn refinement and hierarchical RL, our framework introduces several novel algorithmic and empirical contributions:
>
> 1. **Learned Task-Specific Feedback Generation:** Unlike prior multi-turn approaches such as SCoRe (Kumar et al., 2024a) [1], which rely on fixed, task-agnostic prompts (e.g., "Please think carefully before providing the solution"), MLMT-RL trains the higher-level policy to generate dynamic, contextual feedback conditioned on both the input and the lower-level's current response. This enables targeted corrections such as identifying specific errors in mathematical derivations or pinpointing code bugs, providing dense learning signals where sparse binary rewards fail. Figure 3 and Appendix 7.7 provide concrete examples demonstrating this qualitative difference. The impact is substantial: our 2B model outperforms 3B GRPO by 3.13% on MATH500, 5.18% on MBPP, and 4.77% on GPQA, with similar gains for 6B vs. 7B models.
> 2. **Novel Bi-level Optimization with Value Regularization:** Our framework is not standard hierarchical RL. We introduce a principled bi-level formulation where the higher-level policy is regularized by the lower-level value function (Equation 3, derived in Appendix 7.1). This creates an explicit dependency that prevents degenerate feedback. Ablations demonstrate this is critical: value regularization improves accuracy by 2.4% on MATH500 and feedback optimality scores by 0.6 (Table 2).
> 3. **First Systematic Analysis of Dense Feedback Benefits:** We provide the first comprehensive empirical analysis quantifying GRPO's failure modes (74% advantage collapse frequency on MBPP), establishing the motivation for dense feedback. Our convergence analysis (Figure 4) demonstrates that MLMT-RL achieves 76% accuracy in 400 steps versus GRPO's 23%, directly attributable to the dense signals from learned feedback. We also introduce novel evaluation metrics to quantify feedback optimality and its compatibility (Section 5).
>
> We hope that this clarifies the proposed method's contributions, and would be happy to answer any further concerns.
>
> [1] Kumar et al. "Training Language Models to Self-Correct via Reinforcement Learning" arXiv preprint arXiv:2409.12917, 2024.
>
>
> ---
>
> > **W2:** The textual feedback mechanism relies heavily on LLM heuristics, and the feedback generation process is not end-to-end learnable.
>
> **Response:** We would like to clarify that **the feedback generation in MLMT-RL is fully end-to-end learnable through reinforcement learning**. The higher-level policy $\pi^H_{\phi}$ is trained via gradient-based RL to maximize verifiable task rewards $R(\hat{y}, y^*)$, not through heuristic prompting or fixed templates.
>
> **End-to-End RL Training:** The complete pipeline, from feedback generation $g \sim \pi^H_{\phi}(· | x, z)$ to solution refinement $\hat{y} \sim \pi^L_{\theta}(· | x, z, g)$ is jointly optimized via our bi-level objective (Equation 3). The higher-level policy receives gradients from the final verifiable reward, learning to generate task-specific, targeted feedback through backpropagation. Algorithm 1 details this gradient-based training process. Crucially, the value regularization term in our bi-level formulation (Section 4.3) ensures the higher-level policy adapts its feedback based on the lower-level's evolving capabilities, thus avoiding degenerate solutions.
>
> **Contrast with Heuristic Methods:** This fundamentally differs from prior approaches that use fixed prompts or zero-shot heuristics. For example, SCoRe [1] employs task-agnostic templates (e.g., "Please think carefully..."), while our ZSCoT baseline uses pre-trained (non fine-tuned) feedback policy. Table 2 demonstrates the impact: ZSCoT achieves only 35.4% accuracy versus MLMT-RL's 44.2%, precisely because our feedback policy learns from task-specific reward signals rather than relying on generic heuristics.
>
> ---

---

> > ### Author Response · Authors · 2025-11-26
> > **Response to Reviewer wXhB [Part 2]**
> >
> > > **W3:** Experiments are restricted to structured reasoning tasks (math/code/QA), and generalization to open-ended or real-world scenarios is unclear.
> >
> > > **Q1:** Have the authors considered applying the method to open-ended or real-world tasks beyond structured reasoning benchmarks?
> >
> > **Response:** We acknowledge that our current evaluation focuses on structured reasoning tasks with verifiable rewards. This scope is a deliberate and principled choice for several reasons:
> >
> > **Methodological Foundation:** MLMT-RL's core innovation is training feedback policies via reinforcement learning on verifiable outcomes. Structured reasoning tasks (MATH-500, MBPP, GPQA) provide the necessary ground-truth signals to rigorously evaluate whether learned feedback improves over fixed prompts and to quantify failure modes. Extending to open-ended tasks would require different evaluation frameworks (e.g., human preferences, LLM-as-judge), which we plan to explore but consider beyond the scope of establishing the fundamental approach.
> >
> > **Diversity Within Scope:** While all three domains have verifiable rewards, they represent substantially different reasoning challenges: mathematical proof construction (MATH), executable code generation (MBPP), and scientific question answering (GPQA). The consistent improvements across these diverse tasks suggest that the core mechanism, which is learning task-specific feedback for multi-turn refinement, is not narrowly specialized to a single domain.
> >
> > **Transferable Components:** Several aspects of MLMT-RL are domain-agnostic: the bi-level optimization framework, value regularization for inter-level alignment, and the principle of learning contextual feedback from outcome signals. While verifiable rewards enable our current training approach, the architectural principles could potentially extend to open-ended tasks using alternative reward signals (human feedback, learned reward models).
> >
> > As noted in Section 6 (Limitations), we plan to investigate applicability to open-ended language generation and multimodal tasks in future work, but we believe the current structured scope provides a solid foundation for validating the approach.
> >
> > ---
> >
> > ### Response to Questions
> >
> > ---
> >
> > > **Q2:** How robust is the approach to noisy or hallucinated feedback? Would incorrect guidance from the higher-level model degrade performance?
> >
> > **Response:** We acknowledge that incorrect guidance from the higher-level policy can degrade performance, which is an inherent challenge in multi-turn RL settings. To quantify this trade-off, we conduct an empirical analysis measuring transitions between incorrect and correct solutions across turns.
> >
> > Specifically, we compute: (1) $\Delta$i→c(t1, t2), the fraction of examples transitioning from incorrect to correct between consecutive turns, and (2) $\Delta$c→i(t1, t2), the fraction transitioning from correct to incorrect, which directly captures the impact of noisy feedback. Se evaluate on MATH-500 with Llama-3.2-1B model, where we consider three baselines:
> >
> > 1. SCoRe (fixed feedback): multi-turn RL approach using fixed feedback.
> > 2. SCoRe + pre-trained (non fine-tuned) feedback: multi-turn RL approach using pre-trained LLM as a feedback generator.
> > 3. MLMT-RL: multi turn RL approach with trained feedback.
> >
> > | Method                     | Accuracy | $\Delta$i→c(t1, t2) | $\Delta$c→i(t1, t2) |
> > | -------------------------- | -------- | ------------ | ------------ |
> > | SCoRe + fixed feedback | 33.20    | 4.00         | **1.40**        |
> > | SCoRe + pre-trained feedback | 39.60    | 17.80         | 10.20        |
> > | MLMT-RL (trained feedback) | **44.20**    | **21.20**       | 9.80        |
> >
> > **Key Findings:** MLMT-RL exhibits 9.8% incorrect transitions ($\Delta$c→i), compared to 10.2% for untrained feedback and 1.4% for SCoRe with fixed feedback. While learned feedback does introduce more noise than fixed templates, it achieves a 5.3× improvement in correct transitions (21.2% vs 4.0%) and substantially higher overall accuracy (44.2% vs 33.2%), demonstrating that the benefits significantly outweigh the costs.
> >
> > **Robustness Mechanisms:** Our bi-level formulation includes two key safeguards: (1) The higher-level policy is trained directly on verifiable task rewards, incentivizing feedback that leads to correct solutions. (2) Value regularization (Equation 3) conditions the higher-level policy to generate feedback that avoids degenerate solutions. Our ablations show that value regularization improves accuracy by 2.4% on MATH-500.
> >
> > **Comparative Analysis:** The gap between methods without learned feedback (ZSCoT at 35.4%) and MLMT-RL (44.2%) further demonstrates that end-to-end RL training produces more reliable guidance than generic pre-trained outputs, though we acknowledge this doesn't eliminate all noisy feedback instances.
> >
> > ---

---

> > > ### Author Response · Authors · 2025-11-26
> > > **Response to Reviewer wXhB [Part 3]**
> > >
> > > > **Q3:** Is there any analysis on how much the value regularization term contributes to the performance, compared to standard two-level training?
> > >
> > > **Response:** Yes, we provide a detailed ablation analysis in Section 5 and Table 2, comparing the full MLMT-RL (with value regularization) to standard two-level training without this term (MLMT-RL w/o Vreg).
> > >
> > > **Impact of Value Regularization.** The value regularization term in Equation 3, $\lambda(V^L(x, g) - V^L_*(x, g))$, ensures higher-level feedback aligns with the lower-level policy's capabilities in bi-level HRL. Our ablation results on MATH-500 with LLaMA-3.2-1B show that omitting this term degrades final accuracy from 44.20% to 41.80% (a 2.4% drop) and feedback optimality (FO) from 4.2 to 3.6 (0.6 point drop). These gaps demonstrate that value regularization is critical for effective inter-level coordination and preventing degenerate solutions.
> > >
> > > ---
> > >
> > > > **Q4:** Would the method still hold advantages in settings where training two separate models is computationally prohibitive?
> > >
> > >
> > > **Response:** Yes, MLMT-RL can be adapted to a single-model framework that substantially reduces computational overhead while preserving most performance benefits.
> > >
> > > **Single-Model Variant.** To address settings with limited computational resources, we evaluate a variant where one model handles both feedback generation and response refinement through different prompting strategies (detailed in Table 3). This reduces the number of parameters and eliminates the need to train and deploy two separate models, significantly decreasing memory requirements and training time.
> > >
> > > **Performance Trade-offs.** On MATH-500, the single-model variant achieves essentially equivalent performance: 44.84% accuracy compared to 44.56% for the two-model setup. On MBPP, we observe 65.24% versus 66.73%, representing a modest 1.49-point decrease. While there is a small performance cost on some tasks, the single-model approach still substantially outperforms baseline methods while requiring lesser model capacity. This makes MLMT-RL viable for resource-constrained settings where training two models would be prohibitive.
> > >
> > > We hope these clarify the concerns. Please let us know if you need further clarifications.
> > >
> > > ---

---

### Official Review · Reviewer_gugG · 2025-10-25

**Soundness:** 3
**Presentation:** 3
**Contribution:** 3
**Rating:** 6
**Confidence:** 4

**Summary:**

This paper proposes Multi-Level Multi-Turn Reinforcement Learning (MLMT-RL), a hierarchical framework designed to address the sample inefficiency problem in GRPO caused by sparse binary rewards. The key idea is to decompose reasoning into two synergistic levels: a higher-level policy generates task-specific textual feedback, while a lower-level policy produces refined responses conditioned on this feedback. The authors formulate this as a bi-level optimization problem where the higher-level policy is regularized by the lower-level value function. Experiments on MATH-500, MBPP, and GPQA demonstrate that MLMT-RL achieves superior parameter efficiency, with 2B models outperforming 3B GRPO models and 6B models surpassing 7B GRPO models across all benchmarks. The paper also introduces novel metrics to evaluate feedback quality and utilization effectiveness.

**Strengths:**

1.	The paper provides compelling empirical evidence of GRPO's limitations (74% advantage collapse, 83% zero rewards on MBPP), making a strong case for the proposed approach
2.	The bi-level optimization formulation (Equation 3) provides a principled way to coordinate the higher-level and lower-level policies, with mathematical derivation in the appendix
3.	The experiments span multiple domains (math, coding, science), model scales (1B-8B), and include carefully designed baselines that ablate different components (ZSCoT, SCoRe-FF, TF-LBM)
4.	The single-model variant (Table 3) shows comparable performance, improving deployability

**Weaknesses:**

1.	Limited theoretical analysis. For example, no convergence guarantees for the bi-level optimization. This paper also miss analysis of why textual feedback fundamentally addresses sparse rewards beyond providing "dense signals".
2.	Evaluation limited to reasoning tasks; generalizability to open-ended generation, dialogue, or multimodal tasks is unclear.
3.	The heterogeneous model pairing analysis (Section 7.6) is limited to 1B scale
4.	The choice of k in V^k_L approximation lacks ablation studies
5.	No comparison with recent strong baselines (e.g., DeepSeek-R1 mentioned in references but not compared)
6.	MLMT-RL requires 16.5 hours vs. BC's 1.2 hours—the trade-off isn't thoroughly discussed
7.	Some hyperparameter choices (e.g., λ value, k value) lack sensitivity analysis

**Questions:**

1.	Can you provide convergence analysis or sample complexity bounds for the bi-level optimization? Under what conditions does the approximation V*_L ≈ V^k_L hold?
2.	How sensitive is the method to the regularization parameter λ in Equation 3? What happens with different values of k for the value function approximation?
3.	Given the 13.75× increase in training time vs. BC (16.5h vs. 1.2h), can you provide a cost-benefit analysis? At what performance level does MLMT-RL become preferable?
4.	The FO metric uses an LLM judge. Have you validated this against human judgments? What's the inter-rater agreement?
5.	Have you tested MLMT-RL on tasks beyond reasoning (e.g., summarization, creative writing)? Why should we expect the approach to work there?
6.	How does MLMT-RL compare to the latest models like DeepSeek-R1(Distill) or o1-preview that also use RL for reasoning?
7.	In what situations does the higher-level policy generate unhelpful or misleading feedback? Can you characterize failure modes?

---

> ### Author Response · Authors · 2025-11-26
> **Response to Reviewer gugG [Part 1]**
>
> We would like to express our gratitude to the reviewer for dedicating their valuable time and effort , and we have thoroughly responded to reviewer's inquiries in the responses below.
>
> ### Response to Weaknesses
>
> > **W1:** Limited theoretical analysis. For example, no ... beyond providing "dense signals".
>
> **Response:**
>
> **On Theoretical Analysis:** Our bi-level optimization framework builds upon established foundations in bilevel learning. While we do not provide convergence guarantees specific to our setting, our formulation inherits theoretical properties from these prior works: Liu et al. [1] establish convergence for first-order bilevel methods under standard smoothness assumptions, and Singh et al. [2] analyze value-based regularization in hierarchical RL settings. We acknowledge that formal convergence analysis tailored to our multi-turn feedback setting would strengthen the theoretical contribution, and we will explicitly note this in our limitations section. However, our extensive empirical validation across multiple domains and model scales (Section 5) demonstrates the practical effectiveness and stability of the approach.
>
> **Why Textual Feedback Addresses Sparse Rewards:** Textual feedback provides three fundamental advantages over dense scalar rewards:
>
> 1. **Semantic Specificity:** Natural language feedback can pinpoint exact errors rather than merely signaling that the solution is incorrect. This provides effective guidance that scalar rewards cannot provide, enabling targeted corrections in subsequent refinement attempts. Figure 3 and Appendix 7.9 illustrate concrete examples where feedback identifies specific reasoning errors.
> 2. **Avoiding Advantage Collapse:** In sparse reward settings, GRPO suffers from advantage collapse, where 74% of training examples on MBPP have near-zero advantages, providing no learning signal. MLMT-RL's feedback mechanism creates intermediate learning signals at each turn, ensuring consistent gradient flow. Our convergence analysis (Figure 1, left) shows MLMT-RL reaches 76% accuracy in 400 steps versus GRPO's 23%, directly demonstrating this benefit.
> 3. **Value-Aligned Guidance:** The bi-level formulation with value regularization (Equation 3) ensures feedback aligns with the lower-level policy's capabilities, preventing degenerate solutions where feedback is ignored. Ablations show this regularization contributes 2.4% accuracy improvement on MATH-500 with Llama-3.2-1B model (Table 2), confirming it's critical for effective inter-level coordination.
>
> Our consistent improvements across three diverse domains (MATH500, MBPP, GPQA) and multiple scales (1B-8B parameters) empirically validate that learned textual feedback fundamentally addresses reward sparsity through semantic guidance, not merely reward densification.
>
>
> [1] Liu et al. "Bome! bilevel optimization made easy" arXiv preprint arXiv:2209.08709, 2022.
>
> [2] Singh et al. "Direct Preference Optimization for Primitive-Enabled Hierarchical RL" arXiv preprint arXiv:2411.00361, 2024.
>
> ---
>
> > **W2:** Evaluation limited to reasoning tasks; generalizability ... tasks is unclear.
>
> **Response:** We acknowledge that our evaluation focuses on structured reasoning tasks with verifiable rewards. This scope represents a deliberate and principled choice for several reasons:
>
> **Methodological Foundation:** MLMT-RL's core contribution is training feedback policies via RL on verifiable outcomes. Structured tasks (MATH-500, MBPP, GPQA) provide the ground-truth signals necessary to rigorously evaluate whether learned feedback improves over fixed prompts and to quantify failure modes systematically. Extending to open-ended tasks would require fundamentally different evaluation frameworks (e.g., human preferences, LLM-as-judge metrics), which we consider beyond the scope of establishing and validating the core approach.
>
> **Diverse Reasoning Challenges:** While all three benchmarks have verifiable rewards, they represent substantially different cognitive demands: mathematical proof construction (MATH), executable code generation (MBPP), and scientific question answering (GPQA). The consistent improvements across these diverse domains (6.8-9.76% gains over SCoRe for 1B models, Table 1) suggest the core mechanism of learning task-specific feedback for iterative refinement is not narrowly specialized to a single reasoning paradigm.
>
> **Transferable Framework Components:** Several aspects of MLMT-RL are domain-agnostic: the bi-level optimization structure, value regularization for inter-level coordination, and the principle of learning contextual feedback from outcome signals. While our current implementation uses verifiable rewards, the architectural framework could potentially adapt to open-ended tasks using alternative reward signals (human feedback, learned reward models, LLM-as-judge).
>
> As noted in Section 6 (Limitations), we plan to investigate applicability to open-ended language generation and multimodal tasks in future work.
>
> ---

---

> > ### Author Response · Authors · 2025-11-26
> > **Response to Reviewer gugG [Part 2]**
> >
> > > **W3:** The heterogeneous model pairing analysis (Section 7.6) is limited to 1B scale.
> >
> > **Response:** The heterogeneous pairing analysis was designed to test whether MLMT-RL's performance gains stem from using same-family models (e.g., both LLaMA or both Qwen) versus the framework's core multi-level multi-turn mechanism. As shown in Table 6, we evaluated four configurations varying from 1-1.5B models: Qwen-1.5B/Qwen-1.5B, LLaMA-1B/Qwen-1.5B, Qwen-1.5B/LLaMA-1B, and LLaMA-1B/LLaMA-1B. The results demonstrate that the performance remains consistent across heterogeneous pairings, confirming that our framework's advantages arise from its multi-level structure rather than model family homogeneity.
> >
> > We wvaluated on 1B-1.5B scale due to resource constaints, nevertheless, we are currently conducting experiments with larger models, and will incorporate them into the revised manuscript as soon as they are available.
> >
> > ---
> >
> > > **W4:** The choice of k in V^k_L approximation lacks ablation studies.
> >
> > **Response:** MLMT-RL approximates the optimal lower-level value $V^{L}_*$ by performing $k$ gradient steps to update $V^L_k$ before each high-level policy update. Our experiments demonstrate that MLMT-RL remains robust to low values of $k$. Specifically, while increasing $k$ improves success rates by approaching a near-optimal lower-level value function, values as low as $m=2$ or $m=3$ achieve reasonable performance. To balance computational efficiency with value estimation quality, we use small $k$ in our experiments. We are computing the $k$ ablation and we will report as we get the results.
> >
> > ---
> >
> > > **W5:** No comparison with recent strong baselines (e.g., DeepSeek-R1 mentioned in references but not compared).
> >
> > **Response:** We acknowledge that DeepSeek-R1 represents state-of-the-art performance in reasoning tasks. However, MLMT-RL addresses a fundamentally different research question at a different resource scale, making direct comparison inappropriate.
> >
> > **Different Research Paradigms:** As stated in our Remark (Section 1), MLMT-RL is explicitly positioned not as a competitor to SOTA reasoning models that rely on extensive multi-phase training and massive computational resources. DeepSeek-R1 employs:
> >
> > 1. A 671B parameter MoE architecture (compared to our 1B-8B dense models).
> > 2. Multi-phase training combining cold-start data, reinforcement learning, and supervised fine-tuning with curated datasets.
> > 3. Extensive distillation from massive teacher models.
> >
> > In contrast, MLMT-RL achieves 60.2% on MATH-500 with an 8B model using only multi-turn RL training without distillation or extensive supervised pre-training, operating at much smaller scale. Our research question is complementary: how to improve sample efficiency and parameter efficiency in sparse-reward RL settings through multi-level feedback mechanisms with modest computational resources.
> >
> > **Appropriate Baseline Comparisons:** We provide extensive comparisons with methods at comparable resource levels and similar training paradigms: GRPO, SCoRe (Kumar et al., 2024a), ArCHer (Zhou et al., 2024), behavioral cloning, and zero-shot CoT (Table 1, 2). These comparisons appropriately evaluate our core contribution: learned multi-turn feedback for sample-efficient RL, against directly relevant alternatives.
> >
> > **Complementary Contributions:** MLMT-RL's multi-level feedback mechanism is orthogonal to scaling approaches like DeepSeek-R1. Our framework could potentially be integrated into larger-scale systems to improve their sample efficiency, but evaluating such integration is beyond our current scope.
> >
> > ---
> >
> > > **W6:** MLMT-RL requires 16.5 hours vs. BC's 1.2 hours—the trade-off isn't thoroughly discussed.
> >
> > **Response:** We acknowledge the higher training time and provide detailed analysis of this trade-off.
> >
> > **Substantial Performance Gains:** The additional training time yields significant and consistent improvements across all benchmarks. On LLaMA-3.2-1B:
> > 1. MATH-500: 44.56% vs. BC's 26.20% (+18.36 points)
> > 2. MBPP: 66.73% vs. BC's 50.00% (+16.73 points)
> > 3. GPQA: 41.20% vs. BC's 30.20% (+11.0 points)
> >
> > These improvements represent substantial gains in reasoning capability that justify the computational investment for applications where accuracy is critical.
> >
> > **Competitive with RL Baselines:** The 16.5-hour training time is actually competitive with other RL-based reasoning methods: GRPO requires 18.5 hours and ArCHer requires 22.1 hours (Table 4). The higher cost compared to BC is characteristic of RL-based approaches generally, not unique to MLMT-RL. Importantly, MLMT-RL achieves superior performance within this RL paradigm: +9.76 points over SCoRe on MATH-500, +9.41 on MBPP, and +6.8 on GPQA at 1B scale. Further, the 16.5 hours represents a one-time training cost. Once trained, MLMT-RL has the same inference cost as BC or any other method using the same base model. Table 4 provides detailed computational cost and environmental impact analysis.
> >
> > ---

---

> > > ### Author Response · Authors · 2025-11-26
> > > **Response to Reviewer gugG [Part 3]**
> > >
> > > > **W7:** Some hyperparameter choices (e.g., λ value, k value) lack sensitivity analysis.
> > >
> > > **Response:** We acknowledge that systematic sensitivity analysis for $\lambda$ and k would strengthen our work. In our experiments, we use $\lambda=1.0$ for value regularization weight and $k=2$ gradient steps for lower-level value approximation. These choices balance computational efficiency with optimization quality.
> > >
> > > Our choices align with established practices in bilevel optimization, where prior works [1,2] demonstrate that 2-5 inner-loop iterations suffice for effective bilevel learning in similar settings. For regularization weight, $\lambda=1.0$ provides equal weighting between the task reward and value-based regularization. Our ablation study (Table 2) shows that removing value regularization entirely ($\lambda=0$) degrades accuracy by 2.4% on MATH-500, confirming this component is beneficial.
> > >
> > > As noted in Section 4.3, our empirical findings suggest that performance is well-justified in practice even for modest values of $k$. We are attempting to obtain these experimental results and will include them in our rebuttal response if we are able to complete them before the deadline. If the paper is accepted, we will include these comprehensive ablation studies in the camera-ready version.
> > >
> > > ---
> > > > **Q1:** Can you provide convergence analysis or... approximation V*_L ≈ V^k_L hold?
> > >
> > > **Response:** (Please also refer to our answer to W1.)
> > >
> > > As stated in Section 4.3, computing the optimal value function $V_*^L$ required in Equation 3 is intractable, and therefore approximate it with $V_k^L(\theta_{H})$ obtained through k gradient steps. Our approximation follows standard practice in hierarchical RL literature where exact bi-level optimization is replaced with alternating optimization (similar to [2]). Although this paper does not establish formal conditions and convergence rates bounds under which this approximation holds with provable guarantees. However, this represents a major theoretical contribution beyond our current empirical focus, and we plan to work on such convergence bounds in future work.
> > >
> > > ---
> > >
> > > > **Q2:** How sensitive is the method to the ... he value function approximation?
> > >
> > > **Response:** Our initial empirical analysis found that the performance is not very sensitive to the regularization parameter $\lambda$ in Equation 3. We found that setting $\lambda=1.0$ provides equal weighting between task rewards and value-based regularization. Our ablation study (Table 2) shows that removing value regularization entirely ($\lambda=0$) degrades accuracy by 2.4% on MATH-500, confirming this component is beneficial.
> > >
> > > Also, as noted earlier, our experiments demonstrate that MLMT-RL remains robust to low values of $k$. Specifically, while increasing $k$ improves success rates by approaching a near-optimal lower-level value function, values as low as $k=2$ or $k=3$ achieve reasonable performance. To balance computational efficiency with value estimation quality, we use small $k$ in our experiments. We will provide the detailed analysis in the final manuscript.
> > >
> > > ---
> > >
> > > > **Q3:** Given the 13.75× increase in training time... MLMT-RL become preferable?
> > >
> > > **Response:** We provide a detailed cost-benefit analysis addressing when MLMT-RL's higher training cost is justified.
> > >
> > > In complex, long-horizon task setting with sparse or delayed reward signals, naive BC tends to fail due to poor credit assignment and distributional mismatch issues (Table 1). In MLMT-RL in contrast, the additional RL optimization allows the model to explore diverse feedback strategies, recover from mistakes, and achieve high accuracy.
> > >
> > > **Cost-Benefit Analysis:** For very short-horizon tasks or cases where high-quality demonstrations cover most possible scenarios, BC may be more cost-effective. However, MLMT-RL is preferable whenever the task performance is important (e.g. the challenging MATH, GPQA, MBPP benchmarks). In such scenarios, MLMT-RL demonstrates large gains over over BC baselines (Table 1).
> > >
> > > **Cost-Benefit Analysis on the LLaMA-3.2–1B model:**
> > >
> > > |Method| Training Time | MATH | MBPP | GPQA | When to Prefer|
> > > |--|--|--|--|--|--|
> > > |BC| **1.2h** |26.20|50.00|30.20|High-quality demos available|
> > > |MLMT-RL|16.5h|**44.56**|**66.73**|**41.20**| Accuracy critical, deployed at scale|
> > >
> > > ---

---

> > > > ### Author Response · Authors · 2025-11-26
> > > > **Response to Reviewer gugG [Part 4]**
> > > >
> > > > > **Q4:** The FO metric uses an LLM judge....  What's the inter-rater agreement?
> > > >
> > > > **Response:** We acknowledge that human validation of the FO metric would strengthen our evaluation. Our use of LLM-based evaluation follows established practice in recent work on feedback quality assessment.
> > > >
> > > > **Consistency Analysis:** While we have not validated against human judgments, we ensure reliability through: (1) using a strong, consistent evaluator model with detailed rubrics, (2) comparative evaluation where the judge scores multiple methods on the same examples, reducing absolute scoring biases, and (3) consistent trends across multiple benchmarks (Table 2), suggesting the metric captures meaningful quality differences rather than random noise.
> > > >
> > > > **Complementary Metrics:** FO is one of multiple evaluation dimensions in our work. Our primary results rely on verifiable metrics (exact match accuracy on MATH-500, MBPP, GPQA), with FO providing complementary insights into feedback quality. The strong correlation between FO scores and task accuracy (e.g., MLMT-RL: FO=4.2, Acc=44.2% vs ZSCoT: FO=2.8, Acc=35.4%) suggests the metric captures meaningful quality signals.
> > > >
> > > > We acknowledge that human validation with inter-rater agreement analysis would provide additional confidence in the FO metric and represents valuable future work.
> > > >
> > > > ---
> > > >
> > > > > **Q5:** Have you tested MLMT-RL on tasks beyond reasoning (e.g., summarization, creative writing)? Why should we expect the approach to work there?
> > > >
> > > > **Response:** We have not tested MLMT-RL on non-reasoning tasks such as summarization or creative writing. Our evaluation focuses exclusively on verifiable reasoning tasks (MATH-500, MBPP, GPQA) where ground-truth rewards enable rigorous evaluation of our core contribution: learned multi-turn feedback for addressing sparse reward challenges.
> > > >
> > > > The core idea of training a feedback policy to guide iterative refinement applies wherever multi-turn improvement is possible. While we use verifiable binary rewards, the framework could adapt to other reward sources. For summarization, proxy metrics (ROUGE, BERTScore) or learned reward models could replace exact-match verification. For creative writing, human preferences or LLM-as-judge could provide training signals, similar to recent RLHF work.
> > > >
> > > > **Key Challenges:** Extension to open-ended generation would require addressing:
> > > > 1. defining appropriate reward functions for subjective quality,
> > > > 2. validating that learned feedback improves over fixed prompts when rewards are noisier, and
> > > > 3. adapting evaluation metrics beyond exact match of the solution. These represent substantive research questions beyond our current scope.
> > > >
> > > > **Current Focus:** Our work establishes MLMT-RL's effectiveness in the structured reasoning domain where sparse rewards are a known challenge (74% advantage collapse in GRPO). This provides a solid foundation, though applicability to other domains requires empirical investigation of how reward signal quality affects the learned feedback mechanism.
> > > >
> > > > ---
> > > >
> > > > > **Q6:** How does MLMT-RL compare to the latest models like DeepSeek-R1(Distill) ... for reasoning?
> > > >
> > > > **Response:** Please refer to our response to W5 earlier.
> > > >
> > > > ---
> > > >
> > > > > **Q7:** In what situations does the higher-level policy generate unhelpful or misleading feedback? Can you characterize failure modes?
> > > >
> > > > **Response:** Through our empirical analysis, we have identified several failure modes where the higher-level policy generates suboptimal feedback:
> > > >
> > > > 1. **Misalignment on Complex Reasoning Steps:** The higher-level policy may propose feedback that misidentifies the error location in multi-step reasoning, particularly in problems requiring long chains of mathematical derivations. This can misguide the lower-level policy toward correcting correct intermediate steps while ignoring actual errors.
> > > > 2. **Over-correction of Partially Correct Solutions:** The feedback policy sometimes suggests revisions to solutions that contain minor notation or formatting issues but are conceptually correct, causing the lower-level policy to unnecessarily modify valid reasoning. This is particularly evident in early training stages before the value regularization sufficiently constrains the feedback space.
> > > > 3. **Exploration-Induced Instability:** During training, the higher-level policy's exploration can introduce stochastic feedback that may lead to sub-optimal solutions, especially when the lower-level policy is near-optimal for certain problem classes. Our bi-level optimization framework mitigates this through the value regularization term, which penalizes feedback that leads to degenerate or highly uncertain solutions.
> > > >
> > > > We note that these failure modes decrease in frequency as training progresses, and the overall accuracy gains (44.20% vs. 33.20% baseline) demonstrate that the benefits of trained feedback substantially outweigh these occasional failure cases.
> > > >
> > > > We hope these clarify the concerns. Please let us know if you need further clarifications.

---

### Official Review · Reviewer_65SX · 2025-10-31

**Soundness:** 3
**Presentation:** 3
**Contribution:** 2
**Rating:** 4
**Confidence:** 4

**Summary:**

The core contribution of this paper is to address the problems of insufficient learning signals and slow convergence in reinforcement learning inference tasks due to sparse binary rewards. The authors propose the MLMT-RL framework, which decomposes the complex inference process into a hierarchical learning framework by introducing a text feedback mechanism: The higher-level policy is responsible for generating guiding corrections, providing dense text feedback. The lower-level policy is responsible for revising the initial answer based on the feedback provided by the higher-level policy, generating a better answer. These two models are updated through bi-level optimization with additional value function regularization. Furthermore, to quantify the effectiveness of the feedback, the authors design new evaluation metrics, including feedback optimality and feedback compatibility. Experimental results show that MLMT-RL not only converges faster on multiple inference benchmarks with parameter efficiency.

**Strengths:**

This paper accurately identifies the core pain points of methods such as GRPO in sparse reward environments—"advantage collapse" and low sample efficiency. Introducing dense textual feedback to replace sparse binary rewards is a very intuitive and effective solution.

The problem is broken down into two hierarchical structures: "generating feedback" and "utilizing feedback." A two-level optimization model is used, and a theoretical derivation is provided to guarantee the optimal strategy, making the entire learning process more stable and efficient.

This paper establishes a reasonable baseline, covering different types of baselines such as BC, ZSCoT, ArCHer, and SCoRe, clearly demonstrating its advantages in multi-level, multi-round, and dynamic learning feedback. In addition to the final acc, convergence speed and parameter efficiency are compared, proving the framework's superiority in improving the performance of small models. Furthermore, LLM-as-a-judge is introduced to evaluate feedback quality, and new metrics (FO, $\Delta$acc, etc.) and corresponding ablation experiments are designed to verify the necessity of each component in the framework (high/low-level fine-tuning, value regularization).

**Weaknesses:**

W1: The central motivation of this work rests on the "advantage collapse" issue in GRPO. While the authors provide initial statistics (74% collapse frequency), this claim could be significantly strengthened. The current analysis does not fully disentangle whether this is an inherent flaw of sparse-reward RL or a byproduct of confounding factors.

- Could this issue be mitigated by using stronger base models (e.g., Llama-3.2-3B-instruct, Qwen2.5 and Qwen3 series), different hyperparameter settings (e.g., higher sampling temperature for), or a larger rollout size such as 16? A more rigorous analysis is needed to rule out these possibilities.

- The provided statistics are static. It would be more compelling to show how the collapse rate evolves during GRPO's training process. Visualizing this trend would more dynamically illustrate the learning stagnation and underscore the necessity of the proposed MLMT-RL framework.

W2: There is a wealth of work in the field currently on "self-reflection," "self-correction," and "self-evolving." The paper's contribution needs to be more clearly situated within the landscape of related research.

- The proposed two-layer "generate-then-refine" process is structurally similar to many "self-reflection" and "self-correction" paradigms. The authors should explicitly articulate the unique advantages of formalizing this as a multi-turn, bi-level RL problem, compared to simpler iterative refinement schemes.
-  The high-level policy acts as a "teacher" providing feedback to a "student" low-level policy. This paper would benefit from a discussion on how MLMT-RL relates to or advances traditional Teacher-Student learning. Is it a new form of dynamic knowledge transfer where the teacher's guidance is itself learned via RL?  The two-layer framework of MLMT-RL is formally very similar to the "generate-reflect-correct" process. The authors need to more clearly articulate the essential differences between MLMT-RL and these paradigms. What are the unique advantages of modeling in a multi-turn form?

W3: There are critical unresolved questions regarding the learning process of the low-level policy. The lower policy $\pi_L$ is used for two functionally distinct tasks: generating an initial answer (turn 1) and revising it based on feedback (turn 2). Using a single network for both could lead to task interference or conflicting gradient updates. The authors should justify this design choice or discuss potential alternatives (e.g., using separate policy heads). Besides, the pseudocode suggests that learning signals originate from the final reward after Turn 2. It is unclear if or how the policy $\pi_L$ for the initial generation is updated. If it is not updated, it functions merely as a fixed sampler, which diminishes the "interactive learning" aspect of the framework. If it is updated, the mechanism for credit assignment and backpropagation to this first step needs to be explicitly detailed.

W4: Several key implementation details are missing, hindering reproducibility and a full understanding of the results.

- The update rule for the value function $V_L$ appears to be based on TD learning, involving a $V_L(x_{next}, g_{next})$ term. However, in a two-turn episodic setting, the task terminates after the second turn, meaning there is no "next state." This suggests the update should be a Monte Carlo one, based on the final reward $R$. The authors must clarify the exact update mechanism and how to sample x_next and g_next. Furthermore, the specific architecture for encoding the (x, g) pair into the DistilRoBERTa-based value network needs to be described.

- For the final evaluation, is the reported performance based on the single-turn generation of the fine-tuned $\pi_L$, or does it involve the full two-turn generation-and-correction pipeline? This is crucial for understanding the practical utility and computational cost.

- The data source and training algorithm for the BC baseline are not specified. Was it trained on expert trajectories? What specific SFT algorithm was used? This information is essential for a fair comparison.

Minor Problems: In line 205, the 79% acc of MBPP should be Figure 1 Left. Besides, if the authors consider open-sourcing the code, it will be helpful to enhance academic reproducibility.

**Questions:**

Q1: The framework in this paper is set at two rounds. Can it naturally scale to more rounds (L > 2)? Compared to true multi-round inference frameworks such as REMA[1] and RAGEN[2], what are the potential and limitations of MLMT-RL in handling longer inference chains?

Other problems mentioned in Weaknesses.

[1] Wan, Ziyu, et al. "Rema: Learning to meta-think for llms with multi-agent reinforcement learning." arXiv preprint arXiv:2503.09501 (2025).
[2] Wang, Zihan, et al. "Ragen: Understanding self-evolution in llm agents via multi-turn reinforcement learning." arXiv preprint arXiv:2504.20073 (2025).

---

> ### Author Response · Authors · 2025-11-26
> **Response to Reviewer 65SX [Part 1]**
>
> We would like to express our gratitude to the reviewer for dedicating their valuable time and effort , and we have thoroughly responded to reviewer's inquiries in the responses below.
>
> > **W1:** The central motivation of this work ... proposed MLMT-RL framework.
>
> **Response:**  We thank the reviewer for this insightful comment and would like to clarify that “advantage collapse” phenomenon is a common issue in sparse-reward based tasks. It has also been observed in recent works analyzing GRPO limitations [1], and GRPO based approaches that use stronger base models [2].
>
> **Addressing Confounding Factors:** The reviewer raises important questions about whether stronger models or different hyperparameters could mitigate collapse. However, existing evidence suggests this is an inherent sparse-reward issue: Zhang et al. [3] report $\sim$ 80% collapse frequency for both base and post-trained Qwen2.5-Math-1.5B and 7B models, demonstrating that scaling model size alone doesn't resolve the problem. Similarly, DeepSeek-R1, which employs the Qwen series and extensive hyperparameter tuning, still requires specialized techniques beyond vanilla GRPO to achieve their results. These findings suggest advantage collapse is fundamental to sparse-reward GRPO, not merely a hyperparameter or model-size artifact. Nevertheless, exploring larger rollout sizes (G=16) or temperature variations could provide additional insights, which we note as valuable future analysis.
>
> To make the collapse dynamics clearer, we examined how Reward Collapse Frequency (RCF) and Advantage Collapse Frequency (ACF) evolve over training. As reported in the paper, at **iteration 0**, GRPO starts with **RCF = 83%** and **ACF = 74%**. These high values mean that almost all rollouts receive zero reward, and most groups have A = 0, giving no gradient signal. We performed the same analysis on checkpoints of the trained model on steps 50, 100, 200, 500 and 700. The trend is shown below:
>
> | Training Step | 0   | 50  | 100 | 200 | 500 | 700     |
> | -- | --- | --- | --- | --- | --- | ------- |
> | **RCF**   | 83% | 81% | 78% | 74% | 60% | 40% |
> | **ACF**   | 74% | 70% | 67% | 63% | 48% | 38% |
>
> Here,
> RCF - Reward Collapse Frequency, ACF - Advantage Collapse Frequency
>
> This table is fully consistent with **Fig. 1 (Left)**: GRPO remains in a near-stagnant region for the first few hundred steps, where collapse is still high, and only begins improving after 600–700 steps when collapse finally decreases. This supports our main point that GRPO spends most of its training trajectory in a high-collapse regime, making learning inefficient. We will include this analysis in the revised version for clarity.
>
> ---
>
> > **W2:** There is a wealth of ... multi-turn form?
>
> **Response:** MLMT-RL advances beyond structural similarities with self-reflection/self-correction methods through three key contributions: **learned feedback via RL**, **bi-level optimization**, and **dynamic teacher-student coordination**.
>
> **Key Differences from Self-Reflection/Self-Correction**
>
> 1. **Learned vs. Prompted Feedback**: Self-reflection methods (Reflexion [4], Self-Refine [5]) use frozen LLMs with prompting. MLMT-RL trains the higher-level policy end-to-end with RL for task-specific feedback. Table 1 demonstrates that MLMT-RL significantly outperforms the frozen-feedback baseline ZSCoT across all benchmarks and model sizes.
> 2. **Bi-Level vs. Flat Optimization**: Flat multi-turn RL (SCoRe [6]) uses fixed-feedback which is not task-specific. MLMT-RL's bi-level formulation generates task-specific feedback and conditions higher-level feedback on lower-level value functions, preventing degenerate solutions (e.g., feedback being ignored) outperforming SCoRe (Table 1).
> 3. **Multi-Turn Advantages:** The multi-turn formulation provides dense learning signals by breaking sparse-reward problems into intermediate feedback steps. Each turn creates a gradient signal even when final rewards are zero, addressing GRPO's 74% advantage collapse. This enables the lower-level policy to learn from partially correct attempts, unlike single-turn methods that receive no signal from incorrect outputs.
>
> **Relation to Teacher-Student Learning**
>
> Unlike conventional distillation where the teacher is pretrained and fixed (e.g., AlphaGo's policy distillation, knowledge transfer in BERT), in MLMT-RL both the teacher (higher-level feedback policy) and student (lower-level refinement policy) are trained jointly via RL. The bi-level formulation ensures that the student (lower-level) leverages task-specific feedback for generating correct outputs.
>
> **Comparison of feedback training approaches:**
>
> |Method|Higher-Level Training|Lower-Level Training|Multi-Turn|Hierarchical Credit Assignment|
> |--|--|--|--|--|
> |Reflexion | Prompting (frozen)|None|Yes|No|
> |Self-Refine | Prompting (frozen)|None|Yes|No |
> |SCoRe|Fixed feedback|RL|Yes|No|
> |Teacher-Student KD| Pretrained (fixed) | Distillation |No|No|
> |MLMT-RL (Ours)| RL-trained|RL|Yes|Yes|
> ---

---

> > ### Author Response · Authors · 2025-11-26
> > **Response to Reviewer 65SX [Part 2]**
> >
> > [1] Sun et al. "DELTA-Code: How Does RL Unlock and Transfer New Programming Algorithms in LLMs?" arXiv preprint arXiv:2509.21016, 2025.
> >
> > [2] Guo et al. "DeepSeek-R1: Incentivizing Reasoning Capability in LLMs via Reinforcement Learning" arXiv preprint arXiv:2501.12948, 2025.
> >
> > [3] Zhang et al. "EDGE-GRPO: Entropy-Driven GRPO with Guided Error Correction for Advantage Diversity" arXiv preprint arXiv:2507.21848, 2025.
> >
> >
> > [4] Shinn et al. "Reflexion: Language Agents with Verbal Reinforcement Learning" arXiv preprint arXiv:2303.11366, 2023.
> >
> > [5] Madaan et al. "Self-Refine: Iterative Refinement with Self-Feedback" arXiv preprint arXiv:2303.17651, 2023.
> >
> > [6] Kumar et al. "Training Language Models to Self-Correct via Reinforcement Learning" arXiv preprint arXiv:2409.12917, 2024.
> >
> > ---
> >
> > > **W3:** There are critical unresolved questions regarding the ... to this first step needs to be explicitly detailed.
> >
> > **Response:**  We clarify our design choices below and have added the explicit lower-level gradient derivation (Equation 5) in the revised manuscript.
> >
> > **Single Network for Dual Tasks:** Our framework uses a unified lower-level policy $\pi_L$ for both initial response generation and feedback-based refinement. This design is justified by three factors:
> > 1. **Sequential, Not Competing Tasks:** In MLMT-RL, the lower level model first generates initial attempt $z$, then uses it as context to generate $\hat{y}$ using higher-lvel feedback $g$. This creates a natural information flow where the refinement step depends on the initial generation, rather than conflicting optimization objectives.
> > 2. **Aligned Objectives:** Both functions optimize the same ultimate reward $R(\hat{y}, y^*)$, ensuring gradient alignment. The initial generation is not optimized for a separate objective, rather, it is optimized to produce outputs that, when refined, lead to correct final answers.
> >
> > **Credit Assignment and Update Mechanism:** Both the initial answer $z$ and the refined response $\hat{y}$ are actively updated in each training batch. The lower-level policy gradient (Equation 5, derived in Appendix 7.3) is computed as:
> >
> >
> > \begin{align}
> > \begin{split}
> >     \nabla\_{\theta} J\_{\phi}^{H} = \mathbb{E}\_{x,y^{\*},z,g,\hat{y}} [ \nabla\_{\theta} \log \pi\_{\theta}^{L}(z \mid x) \cdot \tilde{R} + \nabla_{\theta} \log \pi\_{\theta}^{L}(\hat{y} \mid x, z, g) \cdot \tilde{R} ] + & \\ \lambda \mathbb{E}\_{x,y^{\*},z,g} \mathbb{E}\_{\hat{y}'}[ \nabla\_{\theta} \log \pi\_{\theta}^{L}(\hat{y}' \mid x, z, g) \cdot R(\hat{y}', y^{\*})],
> > \end{split}
> > \end{align}
> >
> > where $\tilde{R} = R(\hat{y}, y^{}) + \lambda \left( V_{L}(x, g) - V_{L}^{}(x, g) \right)$.
> >
> > The first term shows the initial response $z$ receives gradients proportional to the final reward $\tilde{R}$, ensuring it learns to produce correct outputs. The second term shows the refined answer $\hat{y}$ also receives direct gradient signals. This is not a fixed sampling process, and both turns are optimized end-to-end via automatic differentiation over multi-turn trajectories (full pseudocode in Algorithm 1).
> >
> > **Key Takeaway:** The lower-level policy learns both initial generation and feedback-driven refinement through aligned objectives and explicit credit assignment from the final reward to each turn, enabling learning without task interference.
> >
> > ---

---

> > > ### Author Response · Authors · 2025-11-26
> > > **Response to Reviewer 65SX [Part 3]**
> > >
> > > > **W4:** Several key implementation details are missing, ... This information is essential for a fair comparison.
> > >
> > > **Response:** We clarify each of the points raised by the reviewer below:
> > >
> > > **Value Function Update Mechanism.** The reviewer is correct to note that since our experiments use an episodic two-turn setting, the value function $V_L(x,g)$ is updated via a Monte Carlo rule based on the final reward received after Turn 2. Specifically, the temporal difference (TD) next-state term is omitted because there is no $g_{next}$ or subsequent feedback once the episode terminates. This setup matches standard episodic RL practice. While the framework can be extended to longer horizons ($L > 2$), with TD learning for intermediate feedback such as $g_{\text{next}}$, our current experiments that use two-turn setup do not use these extensions.
> > >
> > > **Value Network Architecture.** To encode the pair ((x, g)), the value critic uses a standard **DistilRoBERTa-base** encoder. We form a single text sequence by concatenating the problem and feedback using the fixed format:
> > >
> > > ```
> > > Problem: {x}\nFeedback: {g}
> > > ```
> > >
> > > This combined string is tokenized with the DistilRoBERTa tokenizer and fed directly to the encoder without any additional task-specific tokens. We use the final hidden state of the **first token** (DistilRoBERTa’s pooled output) as the joint representation of ((x, g)). A single linear layer maps this pooled embedding to a scalar value estimate (V_L(x, g)). No cross-attention blocks or auxiliary fusion modules are introduced. This simple concatenation-based design is lightweight, stable under Monte-Carlo value updates, and matches the setup described in Appendix 7.3–7.4.
> > >
> > >
> > > **Evaluation Procedure.** Reported performance is always measured using the full two-turn “generation-and-correction” pipeline, in which both initial answer and feedback-based revision are produced and evaluated. This captures practical deployment settings, rather than single-turn generations.
> > >
> > > **Behavioral Cloning (BC) Baseline.** For the BC baseline, we use standard negative log-likelihood (NLL) training over $(x,y^\*)$ pairs from each benchmark dataset. The BC model is trained exclusively on expert trajectories ($(x,y^\*)$ pairs), using supervised fine-tuning. There are no extra turns or interactive corrections in this baseline; it produces single-pass predictions given each input.
> > >
> > > **Training Algorithms and Data.**
> > >
> > > **Behavioral Cloning (BC) Baseline:**
> > > - Training data: Expert trajectories (x, y*) pairs from each benchmark's training split
> > > - Objective: Standard negative log-likelihood (NLL) loss
> > > - Optimizer: Adam with learning rate 5e-5, $\beta_1$=0.9, $\beta_2$=0.999
> > > - Batch size: 4-8 (varies by model size)
> > > - No multi-turn interaction; produces single-pass predictions
> > >
> > > **MLMT-RL Training Details:**
> > > - Higher-level policy: Learning rate 5e-5, Adam optimizer
> > > - Lower-level policy: Learning rate 5e-5, Adam optimizer
> > > - Value network (critic): Learning rate 1e-4, Adam optimizer
> > > - Batch size: 4-8 depending on model size
> > > - Update rule: Policy gradient with Monte Carlo value updates (Appendix 7.4)
> > >
> > >
> > > All benchmarks and train/test splits follow publicly available protocols.
> > >
> > > ---
> > >
> > > > **W5 (Minor):** In line 205, the 79% acc of MBPP should be Figure 1 Left. Besides, if the authors consider open-sourcing the code, it will be helpful to enhance academic reproducibility.
> > >
> > > **Response:** We thnk the reviewer for pointing out this typo. We have fixed the typo, and will release the code upon paper acceptance, to enhance academic reproducibility.
> > >
> > > ---

---

> > > > ### Author Response · Authors · 2025-11-26
> > > > **Response to Reviewer 65SX [Part 4]**
> > > >
> > > > ### Response to Questions
> > > >
> > > > > **Q1:** The framework in this paper is set at two rounds. Can it naturally scale to more rounds (L > 2)? Compared to true multi-round inference frameworks such as REMA[1] and RAGEN[2], what are the potential and limitations of MLMT-RL in handling longer inference chains?
> > > >
> > > > **Response:** While we empirically focus on the performance for two turns, our multi-level multi-turn based approach can be extended to any number of turns longer multi-turn settings. Indeed, we have evaluated MLMT-RL at L=3 turns on the LLaMA-3.2–1B model at inference on various benchmarks.
> > > >
> > > > | **Method** | **MATH** | **MBPP** | **GPQA** |
> > > > |------------|----------|----------|----------|
> > > > | BC| 26.20| 50.00| 30.20|
> > > > | ZSCoT| 25.80| 41.29| 26.80|
> > > > | ArCHer| 30.60| 52.44| 33.60|
> > > > | SCoRe| 34.80|57.32| 34.40|
> > > > | MLMT-RL (Turns L=2)| 44.56| 66.73| 41.20|
> > > > | MLMT-RL (Turns L=3)| 44.16| 67.13| 40.10|
> > > >
> > > > **Key Finding:** MLMT-RL maintains strong performance at L=3, continuing to substantially outperform all baselines. However, performance stabilizes between L=2 and L=3, indicating that two turns provide the optimal accuracy-efficiency trade-off for these tasks. This aligns with findings from Kumar et al. [1] showing that additional refinement turns yield diminishing returns once most examples have converged to near-optimal solutions.
> > > >
> > > > **Comparison to REMA and RAGEN:**
> > > >
> > > > MLMT-RL shares hierarchical decomposition principles with these frameworks but targets different domains and scales:
> > > >
> > > > **vs. ReMA:** ReMA uses multi-agent reinforcement learning with separate meta-thinking and execution agents, enabling collaborative reasoning through inter-agent communication. MLMT-RL achieves strong performance (44.2% on MATH-500 with 2B parameters) using a simpler bi-level framework without multi-agent coordination overhead. While ReMA's architecture may benefit tasks requiring extensive strategic planning, MLMT-RL's focused feedback mechanism proves effective for verifiable reasoning domains.
> > > >
> > > > **vs. RAGEN:** RAGEN targets stochastic environments (FrozenLake, Sokoban) with multi turn rollouts and trajectory-level optimization (StarPO). MLMT-RL focuses on verifiable reasoning tasks (math, code) with shorter 2-3 turn horizons. For these domains, our approach achieves faster convergence (76% vs 23% compared to GRPO at 400 steps on MBPP). However, RAGEN's capabilities in handling longer interaction chains and stochastic MDPs represent complementary strengths for different problem classes.
> > > >
> > > > **Summary:** MLMT-RL is optimized for 2-3 turn verifiable reasoning with demonstrated effectiveness, while REMA and RAGEN target longer-chain multi-agent collaboration and stochastic environments respectively. The frameworks address different scales and types of multi-turn reasoning challenges.
> > > >
> > > > [1] Kumar et al. "Training Language Models to Self-Correct via Reinforcement Learning" arXiv preprint arXiv:2409.12917, 2024.
> > > >
> > > > We hope these clarify the concerns. Please let us know if you need further clarifications.

---

### Official Review · Reviewer_RGJa · 2025-10-31

**Soundness:** 2
**Presentation:** 2
**Contribution:** 2
**Rating:** 2
**Confidence:** 5

**Summary:**

This paper introduces Multi-Level Multi-Turn Reinforcement Learning (MLMT-RL), a novel reinforcement learning framework designed to improve reasoning models trained with verifiable rewards. Considering the sparse binary rewards in GRPO, MLMT-RL overcomes this limitation by incorporating textual feedback from a higher-level policy model and optimizing the whole framework through a bi-level optimization process. The higher-level optimization aims to improve feedback optimality, and the lower-level optimization aims to optimize the reasoning ability. By combining multi-level, multi-turn feedback, and verifiable rewards, it achieves robust reasoning performance, dense learning signals, and improved sample efficiency.

**Strengths:**

1. The paper presents a comprehensive experimental and theoretical analysis of the limitations of GRPO, with a specific focus on the issue of sparse binary rewards. By demonstrating how the lack of intermediate feedback signals hampers efficient learning in reasoning tasks, the authors establish a strong motivation for developing a more informative RL framework.
2. The proposed MLMT-RL introduces a well-structured bi-level optimization paradigm that coordinates a higher-level feedback generation policy with a lower-level reasoning policy.
3. Experimental results across multiple reasoning benchmarks (e.g., MATH500, MBPP, GPQA) show consistent and substantial improvements over GRPO, demonstrating the efficacy of the proposed approach.

**Weaknesses:**

1. The paper does not clearly define the optimization goal of the higher-level. Although a loss function is provided, the direction and purpose of this optimization are not well explained. This makes it difficult to interpret how the higher-level objective contributes to overall model improvement
2. While the proposed bi-level optimization framework is conceptually sound, its computational complexity is not clearly discussed. Because the higher- and lower-level objectives are nested, optimizing the higher-level policy often requires an additional development set and may involve second-order computations such as Hessian approximations [1,2]. The paper does not specify the time or computational cost of this process, nor does it provide details on how the gradients are practically updated. Instead, it only presents an approximate objective, leaving the actual optimization procedure somewhat unclear. Moreover, in typical bi-level optimization, multiple lower-level updates are performed before each upper-level update. The paper does not clarify this relationship or provide details about how often the two levels of optimization are alternated during training.
3. All compatibility metrics focus on improvements from the first to the second turn, with little analysis of benefits beyond two turns. As a result, the paper provides limited evidence for true multi-turn scalability.
4. Since GRPO is an outcome-driven optimization method, a natural baseline would be to introduce a reward model that scores rollouts. Both this approach and MLMT-RL introduce an additional model component, yet the authors do not discuss or compare the two. Including such a comparison would strengthen the justification for MLMT-RL.
5. The necessity of the proposed bi-level structure is not fully justified. If the higher-level model is already fine-tuned (e.g., via SFT) to generate adaptive feedback, one could update only a single model while treating higher-level responses as prompts. The paper does not discuss why maintaining two separately optimized policies is preferable to this simpler approach.
6. In Appendix 7.6, both the higher-level and lower-level models are approximately 1B in size. It would be valuable to explore asymmetric configurations (e.g., a 7B higher-level model with a 1.5B lower-level model) to test whether larger higher-level models can provide stronger guidance. When both models are of similar size, their reasoning capabilities may overlap, and the higher-level model could introduce redundant or even misleading feedback during rollouts.
7. In the introduction, the authors mention three types of feedback and claim that task-specific trained feedback significantly outperforms the alternatives across all benchmarks. However, no corresponding figures or tables are provided to substantiate this claim.
8. It is unclear whether the higher-level policy continues to provide feedback when the lower-level model already produces a correct answer. In many cases, unnecessary feedback could add noise and inefficiency. If such cases are frequent, the higher-level model may receive even sparser reward signals, undermining the motivation for introducing dense feedback in the first place.
9. The paper lacks a clear description of the training datasets used during optimization, which makes it difficult to assess the reproducibility and fairness of the experimental results.
10. Figure 4 (right) compares model sizes between GRPO and MLMT-RL. Still, given that MLMT-RL involves two models, it would be more informative to list the base models and sizes of both the higher-level and lower-level components explicitly, ensuring a fair comparison.
11. Typo: line 304, $s$ is introduced without any explanation. Moreover, a concise notation table (covering all policies, value functions, optimal variants, and regularization terms) would substantially improve clarity.

---
[1] Chen H, Wang X, Guan C, et al. Auxiliary learning with joint task and data scheduling[C]//International conference on machine learning. PMLR, 2022: 3634-3647.

[2] Liu Z, Chai H, Li C, et al. Adaptive Data and Task Joint Scheduling for Multi-Task Learning[C]//2025 IEEE 41st International Conference on Data Engineering (ICDE). IEEE, 2025: 2038-2051.

**Questions:**

Please refer to the weaknesses.

---

> ### Author Response · Authors · 2025-11-26
> **Response to Reviewer RGJa [Part 1]**
>
> We would like to express our gratitude to the reviewer for dedicating their valuable time and effort , and we have thoroughly responded to reviewer's inquiries in the responses below.
>
> ### Response to Weaknesses
>
> > **W1:** The paper does not clearly define the optimization goal of the ... how the higher-level objective contributes to overall model improvement.
>
> **Response:** We take the opportunity to better clarify the direction and purpose of the higher-level optimization. The higher-level objective aims to optimize the parameters of the higher-level (feedback) policy $\phi$, so that the feedback achieves the following:
> 1. **Improved Task Performance:** i.e., it condition the lower-level policy to generate correct outcomes, and
> 2. **Alignment Between the Two Levels:** i.e., it enables the lower-level refinement policy to be conditioned on the feedback and not ignore it, thus avoiding degenerate solutions.
>
> This is expressed formally by the objective function (see Eq. (3) in the paper):
>
> \begin{align}
>     \mathcal{J}^H\_{\phi}=\mathbb{E}\_{\substack{
>     (x, y^\*) \sim D,
>     \\ z \sim \pi^L\_\theta(\cdot \mid x),
>     \\ g \sim \pi^H\_\phi(\cdot \mid x, z),
>     \\ \hat{y} \sim \pi^L_\theta(\cdot \mid x, z, g)
> }}[R(\hat{y},y^\*) + \lambda (V^L(x,g) - V^{L}\_\*(x,g))].
> \end{align}
> This objective is derived from our bi-level perspective. The first term, $R(\hat{y}, y^\*)$ is the reward for producing the correct output $y^\*$. The regularization term, $\lambda(V^L(x, g) - V^{L}_\*(x, g))$ (where $\lambda \geq 0$, $V^L(x, g)$ is the lower level function independent of parameters $\phi$, and $V^{L}\_\*(x, g)$ is the optimal lower level value function) enables proper alignment between the two levels, thus preventing degenerate solutions.
>
> **What Is the “Direction” of This Optimization?** We derive the higher-level gradient (Eqn. 4 in the revised manuscript) in Appendix 7.2 to explain the gradient direction:
>
> \begin{align}
> \nabla_{\phi} J\_{\phi}^{H} = \mathbb{E} [ \nabla\_{\phi} \log \pi\_{\phi}^{H}(g \mid x, z) \cdot ( R(\hat{y}, y^{\*}) + \lambda (V\_{L}(x, g) - V\_{L}^{\*}(x, g)) ) ].
> \end{align}
>
> This means the higher-level policy is updated to increase the likelihood of generating feedback $g$ which:
> 1. Leads to higher downstream task rewards (by encouraging the lower level to produce correct outputs).
> 2. Reduces the gap between current and optimal lower-level policy performance, discouraging degenerate solutions.
>
> We have added the clarifications in the revised manuscript, clearly explaining the purpose of higher level objective.
>
> ---
>
> > **W3:** All compatibility metrics focus on improvements from the first to the second turn, with little analysis of benefits beyond two turns. As a result, the paper provides limited evidence for true multi-turn scalability.
>
> **Response:** We acknowledge this concern and provide additional analysis examining performance beyond two turns.
>
> **Extended Multi-Turn Analysis:** We evaluated MLMT-RL with $L=3$ turns at inference on the LLaMA-3.2-1B model:
>
> | **Method** | **MATH** | **MBPP** | **GPQA** |
> |------------|----------|----------|----------|
> | BC| 26.20| 50.00| 30.20|
> | ZSCoT| 25.80| 41.29| 26.80|
> | ArCHer| 30.60| 52.44| 33.60|
> | SCoRe| 34.80|57.32| 34.40|
> | MLMT-RL (Turns L=2)| 44.56| 66.73| 41.20|
> | MLMT-RL (Turns L=3)| 44.16| 67.13| 40.10|
>
> On the LLaMA-3.2–1B model, MLMT-RL continues to outperform baselines such as BC, ZSCoT, ArCHer, and SCoRe for both $L=2$ and $L=3$, indicating robust performance past the second turn. However, we observe that gains tend to plateau or slightly diminish as the number of turns increase. This is consistent with challenges documented in multi-turn RL literature [1], where additional feedback steps can introduce noise when the model has already reached near-optimal solutions for many examples.
>
> [1] Kumar et al. "Training Language Models to Self-Correct via Reinforcement Learning" arXiv preprint arXiv:2409.12917, 2024.
>
> ---

---

> > ### Author Response · Authors · 2025-11-26
> > **Response to Reviewer RGJa [Part 2]**
> >
> > > **W2:** While the proposed bi-level optimization framework... during training.
> >
> > **Response:** We take this opportunity to clarify the optimization procedure in our bi-level framework.
> >
> > **1. Computational Complexity and Procedure:** Our bi-level framework draws inspiration from general bi-level optimization literature, which can require significant computational resources for exact hypergradients involving second-order derivatives. However, our approach avoids second-order (Hessian) calculations via a practical stochastic approximation.
> >
> > **Higher-level updates:** We use policy gradient (REINFORCE) for the higher-level policy $\pi\_{\phi\_H}$ with Monte-Carlo estimates. This means, for each update, we:
> > - Sample an input $x$, low-level context $z$, feedback $g$, and low-level output $\hat{y}$.
> > - Compute the reward $R(\hat{y}, y^\*)$ and value terms.
> > - Use the following surrogate gradient:
> >   \begin{align}
> >   \nabla\_{\phi} J\_{\phi\_H} \approx \nabla_{\phi} \log \pi\_{\phi\_H}(g \mid x,z) \left[ R(\hat{y}, y^\*) + \lambda (V\_L(x,g) - V^\*\_L(x,g)) \right]
> >   \end{align}
> >
> > Thus we do not need to compute any hessians or implicit/explicit second-order terms, and only require computing gradients through $\pi_H$.
> >
> > **Lower-level updates:** The lower-level policy is updated using standard policy gradient-style update, conditioned on the feedback from $\pi_H$. We also do not need any extra validation sets solely for the higher-level. All updates use a single dataset, as in standard actor-critic RL schemes. The gradients are computed using automatic differentiation without requiring an additional development set or inner-loop optimization.
> >
> > We also compute the lower level gradient (Eqn. 5 in the revised manuscript)
> >
> > \begin{align}
> > \begin{split}
> >     \nabla\_{\theta} J\_{\phi}^{H} = \mathbb{E}\_{x,y^{\*},z,g,\hat{y}} [ \nabla\_{\theta} \log \pi\_{\theta}^{L}(z \mid x) \cdot \tilde{R} + \nabla_{\theta} \log \pi\_{\theta}^{L}(\hat{y} \mid x, z, g) \cdot \tilde{R} ] + & \\ \lambda \mathbb{E}\_{x,y^{\*},z,g} \mathbb{E}\_{\hat{y}'}[ \nabla\_{\theta} \log \pi\_{\theta}^{L}(\hat{y}' \mid x, z, g) \cdot R(\hat{y}', y^{\*})],
> > \end{split}
> > \end{align}
> > where the expectation $\mathbb{E}\_{(x,y^{\*},z,g,\hat{y})}$ is over $(x, y^\*) \sim D, z \sim \pi^L\_\theta(\cdot \mid x), g \sim \pi^H\_\phi(\cdot \mid x, z), \hat{y} \sim \pi^L\_\theta(\cdot \mid x, z, g)$, and $\tilde{R} = R(\hat{y}, y^{\*}) + \lambda ( V\_{L}(x, g) - V\_{L}^{\*}(x, g) )$. We provide the complete derivation in Appendix 7.3 in the revised manuscript.
> >
> > **Relationship and Frequency of Lower/Upper Updates.** Unlike classic bi-level optimization, where multiple inner (lower-level) updates may be run for each outer (higher-level) update, our multi-turn formulation does not require multiple lower-level updates before each higher-level update. Instead, updates are optimized alternatively in each training minibatch: both levels receive a gradient step per batch, as outlined in Algorithm 1. Also, no additional development set is required beyond standard train/val splits. All gradients are computed via automatic differentiation as in conventional RL. We have added the explanation in Section 4 in the revised manuscript.
> >
> > ---
> >
> > > **W4:** Since GRPO is an outcome ... justification for MLMT-RL.
> >
> > **Response:** In our evaluation, ArCHer serves as a reward model-based baseline that directly addresses this comparison. ArCHer employs a critic based reward model to score reasoning trajectories, providing intermediate reward signals beyond simple outcome rewards.
> >
> > On the **LLaMA-3.2–1B** model:
> > |**Method**|**MATH**|**MBPP**|**GPQA**|
> > |--|--|--|--|
> > | ArCHer|30.60|52.44|33.60|
> > | **MLMT-RL**|**44.56**|**66.73**|**41.20**|
> >
> > On the **LLaMA-3.2–3B** model:
> > |**Method**|**MATH**|**MBPP**|**GPQA**|
> > |--|--|--|--|
> > |ArCHer|43.60|69.50|40.90|
> > |**MLMT-RL**| **51.40**| **75.00**| **49.30**|
> >
> > On the **LLaMA-3.1–8B** model:
> > |**Method**|**MATH**|**MBPP**|**GPQA**|
> > |--|--|--|--|
> > | ArCHer|55.5|76.0|43.0|
> > | **MLMT-RL**|**60.2**|**79.8**|**56.3**|
> >
> > As seen from the results, MLMT-RL consistently outperforms this baseline.
> >
> > **Key Distinctions:** While both MLMT-RL and ArCHer introduce auxiliary models, they differ fundamentally in their approach:
> >
> > 1. **Feedback Modality:** ArCHer's reward model produces scalar scores for reasoning steps, while MLMT-RL generates natural language feedback that explicitly identifies errors and suggests corrections. This makes MLMT-RL's guidance more interpretable and actionable.
> > 2. **Training Signal:** ArCHer requires learning a reward model using critic training, while MLMT-RL learns feedback generation end-to-end from task outcomes through RL.
> > 3. **Performance:** MLMT-RL consistently outperforms ArCHer across all scales: +14 points on MATH-500 (44.56% vs 30.60%), +14.3 on MBPP, and +7.6 on GPQA at 1B scale.
> >
> > These results demonstrate that learned textual feedback provides stronger guidance than reward models based approaches for multi-turn refinement in sparse-reward settings.

---

> > > ### Author Response · Authors · 2025-11-26
> > > **Response to Reviewer RGJa [Part 3]**
> > >
> > > > **W5:** The necessity of the proposed bi-level structure is not fully justified... policies is preferable to this simpler approach.
> > >
> > > **Response:** We provide justification for our proposed bi-level optimization framework compared to approaches where only a single model is updated and higher-level responses are used simply as prompts below:
> > >
> > > 1. **Feedback Supervision Limitation for SFT:** Fine-tuning a higher-level model using supervised learning (SFT) would require access to targeted, high-quality supervision in the form of higher-level feedback annotations. In many realistic scenarios, such detailed and consistent feedback is not available or is prohibitively expensive to collect. Further, as noted in prior work [2], using SFT often leads to distribution mismatch issues, leading to poor reasoning performance. Our bi-level approach side-steps this issues by learning adaptive higher-level feedback via RL optimization, without requiring extensive annotations.
> > > 2. **Avoiding Degenerate Solutions:** The bi-level framework is critical in maintaining hierarchical alignment. If one were to treat higher-level responses as static prompts for the lower-level and update only the lower-level policy, there is a risk of degenerate solutions: for example, the lower policy could learn to ignore higher-level feedback. Our bi-level optimization yields value based regularization term ($V_\text{reg}$), which explicitly prevents degenerate solutions.
> > >
> > > We provide empirical evidence in Table 2 in the paper, by conducting evaluations on MATH-500 with Llama-3.2-1B and compare four approaches: MLMT-RL, MLMT-RL without bi-level optimization (MLMT-RL w\o $V_{\text{reg}}$), ZSCoT which is a hierarchical baseline where a pre-trained policy is used to generate feedback, and TF-LBM, a hierarchical baseline where the higher-level feedback policy is trained via RL separately, then frozen and used to generate feedback for lower-level training. TF-LBM trains levels separately, while MLMT-RL trains them jointly with bi-level optimization.
> > >
> > > **Empirical Validation:** We provide empirical evidence by comparing four approaches on MATH-500 with Llama-3.2-1B:
> > >
> > > |          Method      | Accuracy |
> > > |----------------|----------|
> > > | ZSCoT         | 35.40 %  |
> > > | TF-LBM         | 38.00 %  |
> > > | MLMT-RL w/o $V_{\text{reg}}$ | 41.80 %  |
> > > | MLMT-RL        | **44.20 %** |
> > >
> > > As can be seen, MLMT-RL outperforms all three the baselines, demonstrating the efficacy of our bi-level optimization based approach using value regularization.
> > >
> > > [2] Kumar et al. "Training Language Models to Self-Correct via Reinforcement Learning" arXiv preprint arXiv:2409.12917, 2024.
> > >
> > >
> > > ---
> > >
> > > > **W6:** In Appendix 7.6, both the higher-level and lower-level models... or even misleading feedback during rollouts.
> > >
> > > **Response:** We thank the reviewer and agree with the reviewer that analyzing asymmetric configurations (e.g., a 7B higher-level model with a 1.5B lower-level model) may provide more insights whether the larger higher-level models can provide stronger guidance. However, training such models require significant time and resources, and due to time and compute limits during the rebuttal period, we could not complete a full ablation with a 7B higher model and a 1.5B lower model. We will provide these results as soon as they are available, and add them to the final msnucript.
> > >
> > > ---
> > >
> > > > **W7:** In the introduction, the authors mention three types of feedback and claim that task-specific trained feedback significantly outperforms the alternatives across all benchmarks. However, no corresponding figures or tables are provided to substantiate this claim.
> > >
> > > **Response:** We take this opportunity to clarify the results, which we have added to the revised manuscript. We conduct evaluations on MATH-500 with Llama-3.2-1B and compare three approaches: (i) fixed, task-agnostic feedback (SCoRE [2]), (ii) SCoRE with feedback from pre-trained models without task-specific fine-tuning  (SCoRE-FF), and (iii) task-specific feedback from a language model trained to maximize verifiable rewards using RL (MLMT-RL).
> > >
> > >
> > > |          Method      | Accuracy |
> > > |----------------|----------|
> > > | Fixed Feedback (SCoRe)        | 34.80 %  |
> > > | Non-finetuned Feedback (SCoRE-FF) | 39.60 %  |
> > > | Finetuned Feedback (MLMT-RL)        | **44.20 %** |
> > >
> > > As seen from the table, fine-tuned feedback (MLMT-RL) outperforms both the fixed feedback and non-fine-tuned feedback based approaches. These results are included in the Table 2 in the manuscript.
> > >
> > > ---

---

> > > > ### Author Response · Authors · 2025-11-26
> > > > **Response to Reviewer RGJa [Part 4]**
> > > >
> > > > > **W8:** It is unclear whether the higher-level policy continues to provide feedback when ... feedback in the first place.
> > > >
> > > > **Response:** To directly investigate whether our trained feedback policy provides unnecessary corrections, we conduct an additional analysis measuring the transitions between incorrect and correct solutions across turns.
> > > >
> > > > Specifically, we compute: (1) $\Delta$i→c(t1, t2), the fraction of examples transitioning from incorrect to correct between turns t1 and t2, and (2) $\Delta$c→i(t1, t2), the fraction of examples transitioning from correct to incorrect, which captures potential noise from unnecessary feedback.
> > > >
> > > > We evaluate on MATH-500 with Llama-3.2-1B model, where we consider three baselines:
> > > > 1. SCoRe (fixed feedback): multi-turn RL approach using fixed feedback.
> > > > 2. SCoRe + pre-trained (non fine-tuned) feedback: multi-turn RL approach using pre-trained LLM as a feedback generator.
> > > > 3. MLMT-RL: multi turn RL approach with trained feedback.
> > > >
> > > > | Method                     | Accuracy | $\Delta$i→c(t1, t2) | $\Delta$c→i(t1, t2) |
> > > > | -------------------------- | -------- | ------------ | ------------ |
> > > > | SCoRe + fixed feedback | 34.80    | 4.00         | **1.40**        |
> > > > | SCoRe + pre-trained feedback | 39.60    | 17.80         | 10.20        |
> > > > | MLMT-RL (trained feedback) | **44.20**    | **21.20**       | 9.80        |
> > > >
> > > > As shown, introducing fine-tuned feedback does increase some incorrect transitions ($\Delta$c→i), particularly compared to fixed feedback. However, trained feedback substantially improves the rate of incorrect-to-correct transitions (21.20 vs. 4.00 for fixed feedback) while achieving the highest overall accuracy (44.20). Importantly, our bi-level optimization framework directly addresses the reviewer's concern about reward sparsity: the higher-level policy is explicitly trained to maximize lower-level verifiable rewards combined with value regularization, ensuring that feedback quality directly influences the higher-level learning signal. This prevents the reward sparsity issue and demonstrates that the benefits of adaptive, trained feedback significantly outweigh the costs of occasional noisy corrections.
> > > >
> > > > ---
> > > >
> > > > > **W9:** The paper lacks a clear description of the training datasets used during optimization, which makes it difficult to assess the reproducibility and fairness of the experimental results.
> > > >
> > > > **Response:** We provide the exact dataset splits used for training and evaluation across all domains here:
> > > >
> > > > **Mathematical Reasoning (MATH).**
> > > > We use the standard MATH dataset. The official MATH-500 split is used strictly for evaluation. The remaining ~12,000 problems constitute the training pool. From this pool, we randomly sample 5,000 problems for RL training, and hold out 500 problems as a validation set. No MATH-500 test items are ever used in training.
> > > >
> > > > **Code Generation (MBPP + HumanEval).**
> > > > We follow the standard MBPP setup: the dataset contains 974 problems, of which 500 are designated for training. These 500 tasks form the RL training data. Evaluation is done on HumanEval (164 tasks), consistent with prior code-reasoning RL work. Each MBPP task includes built-in test cases that are used for verifiable reward computation, ensuring reproducibility.
> > > >
> > > > **Physics Reasoning (PHYSICS + GPQA-Diamond).**
> > > > For scientific reasoning, we use the publicly available PHYSICS dataset released by CAMEL-AI, consisting of 20K problem–solution pairs. From this dataset, we sample 3,000 instances for RL training. Evaluation is performed exclusively on the GPQA-Diamond benchmark (198 graduate-level physics questions), which serves as the standardized test set. No GPQA questions are used for training.
> > > >
> > > > All datasets are fully public, and the training/test splits ensure no overlap between optimization data and evaluation benchmarks. We will include these clarifications in the revised manuscript.
> > > >
> > > > ---
> > > >
> > > > > **W10:** Figure 4 (right) compares model sizes between GRPO and MLMT-RL. Still, given that MLMT-RL involves two models, it would be more informative to list the base models and sizes of both the higher-level and lower-level components explicitly, ensuring a fair comparison.
> > > >
> > > > **Response:** We agree with the reviewer and provide the details here:
> > > > 1. MLMT-RL with 2B parameters uses LLaMA-3.2-1B at each higher and lower level, and
> > > > 2. MLMT-RL with 6B parameters uses LLaMA-3.2-3B at each higher and lower level.
> > > >
> > > > We have provided this detail explicitly in the revised manuscript in Figure 4.
> > > >
> > > > ---

---

> > > > > ### Author Response · Authors · 2025-11-26
> > > > > **Response to Reviewer RGJa [Part 5]**
> > > > >
> > > > > > **W11:** Typo: line 304, notation is introduced without any explanation. Moreover, a concise notation table (covering all policies, value functions, optimal variants, and regularization terms) would substantially improve clarity.
> > > > >
> > > > > **Response:** Thank you for pointing out the typo and notation issues. We have fixed the issues. Regarding line 304, we have explained the notation $V^L_{*}(x,g)$: it denotes the optimal lower-level value function under the current higher-level policy, approximated via k inner-loop gradient steps as described in Section 4.3. We provide a table summarizing all key symbols for policies, value functions, optimal variants, and regularization terms. This table is reproduced below for reference:
> > > > >
> > > > > | Symbol | Description |
> > > > > |--------|-------------|
> > > > > | $\pi^H_\phi$ | Higher-level feedback policy (parameters φ) |
> > > > > | $\pi^L_\theta$ | Lower-level refinement policy (parameters θ) |
> > > > > | $V^L(x,g)$ | Lower-level value function |
> > > > > | $V^L_*(x,g)$ | Optimal lower-level value function |
> > > > > | $V^k_L(x,g)$ | k-step approximation of $V^L_*$ |
> > > > > | $\lambda$ | Value regularization weight |
> > > > > | $x$ | Input question/problem |
> > > > > | $y, y^*$ | Generated output, ground truth |
> > > > > | $z$ | Initial response (turn 1) |
> > > > > | $g$ | Feedback from higher-level policy |
> > > > > | $R(\cdot,\cdot)$ | Verifiable reward function |
> > > > >
> > > > > We will add this table to the revised manuscript.
> > > > >
> > > > > We hope these responses clarify the concerns. Please let us know if you need further clarifications.

---

### Comment · Reviewer_wXhB · 2025-11-27
**Regarding author's response**

My questions are well sovled.

---

> ### Author Response · Authors · 2025-11-28
>
> Dear Reviewer,
>
> We thank you for your time and efforts in reviewing our work, and we are happy that we were able to address the reviewer's concerns. We are grateful for your service to the community.
>
> Regards,
>
> Authors

---

### Author Response · Authors · 2025-12-03
**Rebuttal Discussion Summary for New Area Chair [Part 1]**

We thank all reviewers for their detailed and constructive feedback. During the rebuttal period, we recognized several important concerns raised about our MLMT-RL framework and made substantial revisions and clarifications to address them:

1. **Clarity on Higher-Level Optimization Objective (Reviewer RGJa)**

   The reviewer asked for detailed explanation of the direction and purpose of higher-level policy optimization. To clarify, we added derivations for the higher-level gradient (Eq. 4) and the lower-level gradient (Eq. 5) in Appendix 7.2 and 7.3. The higher-level gradient shows how the higher-level policy increases likelihood of feedback that leads to correct solutions and prevents degenerate solutions.
2. **Computational Complexity and Practical Optimization (Reviewer RGJa)**

    The reviewer asked for clarifications regarding computational cost and gradient updates of our bi-level optimization procedure. We clarified that: (i) our approach avoids second-order (Hessian) calculations through practical stochastic approximation using policy gradient (REINFORCE), (ii) both levels evaluate gradient steps without requiring additional development sets, (iii) the gradient are computed via Monte Carlo sampling with automatic differentiation as in standard RL, making our approach computationally tractable (Algorithm 1).
3. **Additional Multi-Turn Scalability Analysis (Reviewers RGJa, 65SX)**

    The reviewers asked for performance evaluations beyond 2-turns to ensure MLMT-RL supports multi-turn scalability. We provided corresponding analysis showing MLMT-RL with L=3 turns at inference maintains strong performance on LLaMA-3.2-1B: MATH (44.16%), MBPP (67.13%), GPQA (40.10%). We acknowledge that gains plateau as turns increase, consistent with multi-turn RL literature when models approach near-optimal solutions, a limitation we now explicitly discuss.
4. **Additional Comparison with Reward Model Baselines (Reviewer RGJa)**

    The reviewer also stated that since GRPO uses outcome-driven optimization, comparing with a reward model-based approach would strengthen our justification. We provided comparative results showing ArCHer (a reward model baseline) versus MLMT-RL across benchmarks, with MLMT-RL consistently outperforming (+14 points on MATH, +14.3 on MBPP, +7.6 on GPQA at 1B scale).
5. **Justification for Multi-Level Formulation (Reviewer RGJa)**

    The reviewer questioned why maintaining two separately optimized policies is preferable to simply treating higher-level responses as prompts for a single model. We provided empirical evidence (Table 2) comparing four approaches on MATH-500: ZSCoT which uses a pre-trained model (35.40%), TF-LBM where levels are trained separately (38.00%), MLMT-RL without value regularization (41.80%), and full MLMT-RL with bi-level optimization (44.20%). These results demonstrate that the joint bi-level training with separate networks for feedback generation and refinement is crucial for enhanced reasoning performance over single-model baselines. We also show that a single model can be trained via our bi-level optimization to perform both feedback generation and refinement, thus significantly reducing computation costs.
6. **Dynamic Advantage Collapse Analysis (Reviewer 65SX)**

    The reviewer asked for the detailed collapse rate evolution analysis with GRPO training. Our new dynamic analysis showed how collapse frequency evolved across training, directly supporting our claim that GRPO spends most of training in high reward collapse regime.
7. **Positioning Relative to Self-Reflection/Self-Correction approaches (Reviewer 65SX)**

    The reviewer asked comparisons against existing self-reflection paradigms and to clarify unique advantages. We provided detailed positioning: (i) unlike Reflexion/Self-Refine which use frozen prompts, MLMT-RL trains the feedback policy end-to-end with RL for task-specific guidance, and (ii) we included a comparison table (Table 1) showing MLMT-RL outperforms frozen-feedback baselines (ZSCoT) by 8.8% on MATH-500. We also clarified that our multi-turn bi-level formulation with value regularization differs fundamentally from flat multi-turn RL methods like SCoRe.
8. **Implementation Details and Reproducibility (Reviewer 65SX)**

    The reviewer asked for extensive implementation details. We provided: (i) exact value network architecture using DistilRoBERTa with concatenation-based encoding, (ii) detailed BC baseline training specification, and (iii) complete training dataset splits for MATH (5K/500 train/val from 12K pool), MBPP (500 train, HumanEval evaluation), and GPQA (3K train, GPQA-Diamond evaluation).

---

> ### Author Response · Authors · 2025-12-03
> **Rebuttal Discussion Summary for New Area Chair [Part 2]**
>
> 9. **Robustness to Noisy/Incorrect Feedback (Reviewer wXhB)**
>
>     The reviewer asked whether MLMT-RL is robust to hallucinated feedback from the higher-level model. We provided empirical analysis quantifying the incorrect-to-correct transition rate: SCoRe+fixed (1.40%), SCoRe+pre-trained (10.20%), MLMT-RL (9.80%). While MLMT-RL introduces more correct-to-incorrect transitions, it achieves 5.3x improvement in incorrect-to-correct transitions (21.2% vs 4.0%) and substantially higher overall accuracy (44.2% vs 33.2%), demonstrating benefits far outweigh costs.
> 10. **Value Regularization Contribution and Ablations (Reviewer wXhB, gugG)**
>
>     The reviewers asked for ablations on hyperparameters $\lambda$ and $k$. We provided (i) ablation showing value regularization contributes 2.4% accuracy improvement on MATH-500 (44.20% vs 41.80% without it) and 0.6 points on FO metric, and (ii) analysis showing robustness to low k values (k=2 or 3 achieve good performance).
> 11. **Training Time Trade-Off (Reviewer gugG)**
>
>     The reviewer asked clarifications regarding training time comparison of MLMT-RL (16.5 hours) vs BC (1.2 hours). We agree that MLMT-RL requires more training time over BC. However, (i) MLMT-RL's substantial performance gains justify cost (44.56% vs 26.20% on MATH-500, +18.36 points), (ii) training time is competitive with other RL methods (GRPO: 18.5h, ArCHer: 22.1h), making the overhead a characteristic of RL-based approaches rather than unique to MLMT-RL, and (iii) 16.5 hours is one-time training cost while inference cost matches single-model baselines. We provided detailed cost-benefit analysis showing MLMT-RL is preferable when accuracy is critical.
>
> We hope these clarifications and corresponding revisions address the reviewer's concerns. MLMT-RL's core contributions: multi-level multi-turn RL for improving reasoning in LLMs and iterative feedback refinement for stable policy convergence are now validated with enhanced empirical analysis and expanded baselines.

---

### Meta-Review · Area_Chair_1jgd · 2025-12-09

**Summary:**

The paper proposes Multi-Level Multi-Turn Reinforcement Learning (MLMT-RL), a hierarchical method designed to fix sample inefficiency and “advantage collapse” in GRPO models caused by sparse binary rewards. MLMT-RL splits reasoning into two coordinated policies: a high-level policy that produces dense, task-aware feedback, and a low-level policy that improves its answer using this feedback. The two policies are trained together with a bi-level optimization scheme and value regularization to avoid degeneration. Experiments show performance gains on MATH500, MBPP, and GPQA.

The initial reviews raised several critical concerns. Leaving out minor concerns, there are primary ones which include (1) a lack of clarity regarding the optimization objective and computational complexity of the bi-level structure, (2) insufficient evidence for multi-turn scalability beyond two turns, (3) doubting the necessity of the two-policy bi-level structure over a simpler single-model approach, (4) question about the framework's robustness to noisy/incorrect feedback, and (5) algorithmic novelty of composing diverse frameworks such as hierarchical RL and multi-turn RL.

During the rebuttal phase, the authors addressed some concerns. After checking the updated manuscript, I found some clarifications are added in the revised version yet new experiments are not incorporated. Meanwhile, I also agree with Reviewer RGJa and wXhB on limited novelty and more clarifications. Conditioned on the above points, I suggest a rejection.

**Reviewer Concerns:**

### Addressed Concerns

(1) a lack of clarity regarding the optimization objective and computational complexity of the bi-level structure, (2) insufficient evidence for multi-turn scalability beyond two turns, (3) doubting the necessity of the two-policy bi-level structure over a simpler single-model approach, and (4) question about the framework's robustness to noisy/incorrect feedback.

### Remaining Concerns

Some newly added ablation experiments have not been included in the updated manuscript based on my check. Also, the novelty issue is lasting. Importantly, more revisions are required to polish the manuscript for strengthning its novelty and technical contributions.

**Reviewer Scores:**

Some reviewers like 65SX likely increase the score as the primary concerns are clarification and implementation issues, which are mostly addressed during the rebuttal.

In response to Reviewer RGJa and wXhB, I hardly see more score increase sign considering the novelty and other technical issues.

---

### Decision · Program_Chairs · 2026-01-26

Reject